# Anle138b binds predominantly to the central cavity in lipidic Aβ₄₀ fibrils and modulates fibril formation

Mookyoung Han [1], Benedikt Frieg [2], Dirk Matthes[3], Andrei Leonov[1,4], Sergey Ryazanov[1,4], Karin Giller[1], Evgeny Nimerovsky[1], Marianna Stampolaki[1], Kai Xue[1], Kerstin Overkamp[1], Christian Dienemann [5], Dietmar Riedel[6], Armin Giese[4], Stefan Becker [1], Bert L. de Groot [3], Gunnar F. Schröder [2,7], Loren B. Andreas [1] ✉ & Christian Griesinger [1,8] ✉

Alzheimer's disease is a specific neurodegenerative disorder, distinct from normal aging, with a growing unmet medical need. It is characterized by the accumulation of amyloid plaques in the brain, primarily consisting of amyloid beta (Aβ) fibrils. Therapeutic antibodies can slow down the disease, but are associated with potential severe side effects, motivating the development of small molecules to halt disease progression. This study investigates the interaction between the clinical drug candidate small molecule anle138b and lipidic Aβ₄₀ fibrils of type 1 (L1). L1 fibrils were previously shown to closely resemble fibrils from Alzheimer's patients. Using high-resolution structural biology techniques, including cryo-electron microscopy (cryo-EM), nuclear magnetic resonance (NMR) spectroscopy enhanced by dynamic nuclear polarization (DNP), and molecular dynamics (MD) simulations, we find that anle138b selectively binds to a cavity within the fibril. This structural insight provides a deeper understanding of a potential drug-binding mechanism at the atomic level and may inform the development of therapies and diagnostic approaches. In addition, anle138b reduces fibril formation in the presence of lipids by approximately 75%. This may suggest a mechanistic connection to its previously reported activity in animal models of Alzheimer's disease.

Alzheimer's disease (AD) is a chronic neurodegenerative disorder that causes brain damage and cognitive decline and remains a significant challenge with no currently approved cure. The accumulation of amyloid beta (Aβ) is considered a primary pathological hallmark of the disease[1,2], and as the condition progresses, tau protein aggregation[3,4]

and hyperphosphorylation[5-7] are observed. Aβ₄₀ and Aβ₄₂ are the two main Aβ isoforms, with the 2-residue longer peptide, Aβ₄₂, being more prone to aggregation. Aggregates of both Aβ types and tau contribute to the formation of amyloid plaques and neurofibrillary tangles (NFTs) in the central nervous system[8]. Thus, understanding the structural

[1]Department of NMR-Based Structural Biology, Max Planck Institute for Multidisciplinary Sciences, Göttingen, Germany. [2]Ernst-Ruska Centre for Microscopy and Spectroscopy with Electrons, ER-C-3 Structural Biology, Forschungszentrum Jülich, Jülich, Germany. [3]Department of Theoretical and Computational Biophysics, Max Planck Institute for Multidisciplinary Sciences, Göttingen, Germany. [4]MODAG GmbH, Mikroforum Ring 3, Wendelsheim, Germany. [5]Department of Molecular Biology, Max Planck Institute for Multidisciplinary Sciences, Göttingen, Germany. [6]Laboratory of Electron Microscopy, Max-Planck-Institute for Multidisciplinary Sciences, Göttingen, Germany. [7]Physics Department, Heinrich Heine University Düsseldorf, Düsseldorf, Germany. [8]Cluster of Excellence "Multiscale Bioimaging: From Molecular Machines to Networks of Excitable Cells" (MBExC), University of Göttingen, Göttingen, Germany. ✉e-mail: land@mpinat.mpg.de; cigr@mpinat.mpg.de

characteristics of Aβ aggregates is essential for developing effective therapeutic strategies against this debilitating disease.

Currently, antibody therapies targeting Aβ aggregates have shown promise in slowing cognitive decline associated with disease progression. However, these antibodies have also been reported to cause adverse effects when administered at high doses[3]. Therefore, recent research has increasingly focused on the discovery of small molecules that can inhibit amyloid-beta aggregation while effectively crossing the blood-brain barrier.

Building on previous research, multiple studies have used in vitro systems and patient-derived samples to investigate different Aβ fibril polymorphic structures using techniques such as cryo-electron microscopy (cryo-EM) and solid-state nuclear magnetic resonance (ssNMR). Among these polymorphs, the type 1 lipidic Aβ$_{40}$ fibril[9–11] is predominantly observed when the protein is allowed to aggregate in the presence of negatively charged DMPG lipid vesicles (lipid-to-protein molar ratio (LPR) = 30:1) at pH 6.5 and 37 °C, as reported previously[9]. This fibril type, henceforth referred to as the L1 Aβ$_{40}$ fibril, has been shown to resemble the pathological structures observed in the brains of patients[10,11].

The L1 Aβ$_{40}$ fibrils consist of two protofilaments that share an identical fold and are symmetrically twisted together, forming a double-protofilament helical structure. These fibrils also maintain an approximate pseudo-2$_1$ screw symmetry, which contributes to their structural stability. Each monomer is composed of three β-strands, with a long transition region (Ala21-Gly33, hereafter referred to as the loop region) located between the 2$^{nd}$ β-strand (β2: Lys16-Phe20) and the 3$^{rd}$ β-strand (β3: Leu34-Val36). A cavity is formed at the interface between the protofilaments arranged with 2 fold symmetry, with the loop regions from each monomer contributing to cavity formation. This cavity is centrally located along the fibril axis and is hereafter referred to as the central cavity (Supplementary Fig. 2b, c). The L1 Aβ$_{40}$ fibril is formed in the presence of DMPG liposomes, where lipid acyl chains interact with hydrophobic surfaces of the fibril (L17−F19 and A30−V36) through non-covalent interactions[9].

In particular, the β-sheet architecture and protofilament interactions of L1 Aβ$_{40}$ fibrils are similar to those in patient-derived samples. These structural parallels indicate that L1 Aβ$_{40}$ fibrils represent key features of AD pathology and thus serve as a valuable model for developing therapeutic strategies targeting Aβ aggregation. This study aims to use an in vitro system to simulate disease-relevant conditions and assess the effects of a clinically relevant small molecule candidate on Aβ$_{40}$ fibrils, offering insights into disease-modifying mechanisms.

Recent advances in small molecule therapeutics have highlighted anle138b [3-(1,3-benzodioxol-5-yl)−5-(3-bromophenyl)−1H-pyrazole][12] as a promising drug candidate capable of modulating amyloid aggregation. Anle138b is a hydrophobic small molecule that can cross the blood-brain barrier, showing potential as a therapeutic agent for neurodegenerative diseases. It has demonstrated disease-modifying effects in various neurodegenerative disease models[12–19], including an AD mouse model (APP/PS1), by reducing Aβ$_{42}$ plaque formation and protecting mitochondrial integrity[17]. In addition, it targets multiple proteins implicated in neurodegenerative diseases, including tau (AD), α-synuclein (Parkinson's, MSA), and prion protein (CJD)[12–19]. This broad spectrum of activity suggests that the mechanism by which anle138b inhibits Aβ$_{42}$ accumulation may extend to Aβ$_{40}$, making its effects on Aβ$_{40}$ fibrils a significant research focus. Anle138b has completed Phase I clinical trials in healthy volunteers[19] and Parkinson's disease patients (NCT04685265) and is currently being evaluated in Phase II clinical trials for multiple system atrophy (MSA) (NCT06568237).

The present study investigates how anle138b binds to and affects the L1 Aβ$_{40}$ fibrils. The structural and functional effects of anle138b treatment are analyzed under two conditions: pre-aggregation (pre-treatment) and post-fibril formation (post-treatment). By utilizing solid-state nuclear magnetic resonance (ssNMR), cryo-EM, and molecular dynamics (MD) simulations, we aim to gain a detailed understanding of the molecular interactions between anle138b and L1 Aβ$_{40}$ fibrils.

## Results
### Anle138b inhibits L1 Aβ$_{40}$ fibril formation
To assess the effect of anle138b on L1 Aβ$_{40}$ fibril growth, Aβ$_{40}$ monomers were aggregated in the presence of DMPG liposomes containing anle138b at three different Small Molecule-to-Protein molar Ratios (SMPRs; anle138b: Aβ$_{40}$) of 0, 0.6, or 1.2. Thioflavin T (ThT) fluorescence and circular dichroism (CD) analyses revealed a concentration-dependent inhibition of fibril formation by anle138b (Fig. 1a, Supplementary Fig. 1). CD spectra showed β-sheet signatures under all conditions at the beginning of incubation after mixing Aβ$_{40}$ with DMPG liposomes (Supplementary Fig. 1b), indicating similar initial secondary structures regardless of the presence of anle138b. After 48 hours of incubation, a strong β-sheet signal was retained in the control sample (SMPR = 0), whereas β-sheet formation was reduced in samples treated with anle138b (SMPRs = 0.6 and 1.2). Fibril formation was most reduced at SMPR 1.2, notably indicating a dose dependence (Supplementary Fig. 1a, c). These results were further supported by two complementary approaches performed on a dedicated fibril sample (pre-treatment, post-treatment, and control fibril): 1D ($^1$H)$^{15}$N CP spectrum signal intensity, and quantification of fibrils by negative stain EM and CD spectroscopy (Fig. 1b, c, Supplementary Fig. 3). All three methods consistently indicated a dose-dependent inhibition of fibril formation by anle138b.

To evaluate the effect of anle138b at different stages of fibril formation, we compared three experimental conditions: 1) pre-treatment (SMPR = 1.2, with anle138b added before initiating fibril formation), 2) post-treatment (SMPR = 1.2, with anle138b applied after fibril formation was completed), and 3) control (fibril without anle138b) (Supplementary Fig. 2a). Pre-formed fibrils used in the post-treatment condition were generated with DMPG liposomes under identical conditions to the control fibrils. Fibril formation under the pre-treatment condition was reduced by approximately 75% compared to the control (Fig. 1c). This observation was supported by analysis of the supernatant following ultracentrifugation: CD and ThT fluorescence spectra showed that β-strand-like species remained in the supernatant under pre-treatment conditions, indicating the presence of soluble, non-fibrillar Aβ$_{40}$ aggregates (Supplementary Fig. 3c, d). In contrast, these species were absent in the control or post-treatment samples, where nearly all Aβ$_{40}$ sedimented as fibrils (Supplementary Fig. 3c, d), i.e., post-treatment resulted in no significant change in fibril quantity (Fig. 1c, Supplementary Fig. 3b). The observed reduction in ThT fluorescence intensity in the post-treatment sample (the fibril treated with anle138b after their formation) is likely attributable to competitive binding between anle138b and ThT at shared or nearby fibril binding sites. Thus, the fluorescence intensity cannot be used as a measure of fibril quantity, and complementary readouts suggest that the fibril quantity has not decreased significantly (Fig. 1b, c, Supplementary Fig. 3b). This interpretation is consistent with previously reported interactions between small molecules and amyloid fibrils[20,21].

In addition, CD spectroscopy and negative stain EM imaging confirmed that fibrils remained structurally unchanged after post-treatment with anle138b, even after 96 hours of incubation at 37 °C (Supplementary Fig. 4). These findings indicate that mature fibrils retain their structural integrity in the presence of anle138b.

### Structural preservation and local effects of anle138b on L1 Aβ$_{40}$ Fibrils
To determine whether the inhibition of fibril formation by anle138b is accompanied by structural changes, we analyzed the fibrils using cryo-EM and ssNMR. Consistent with previous findings, the ssNMR data[9] indicate that L1 Aβ$_{40}$ fibrils represent the predominant species of the

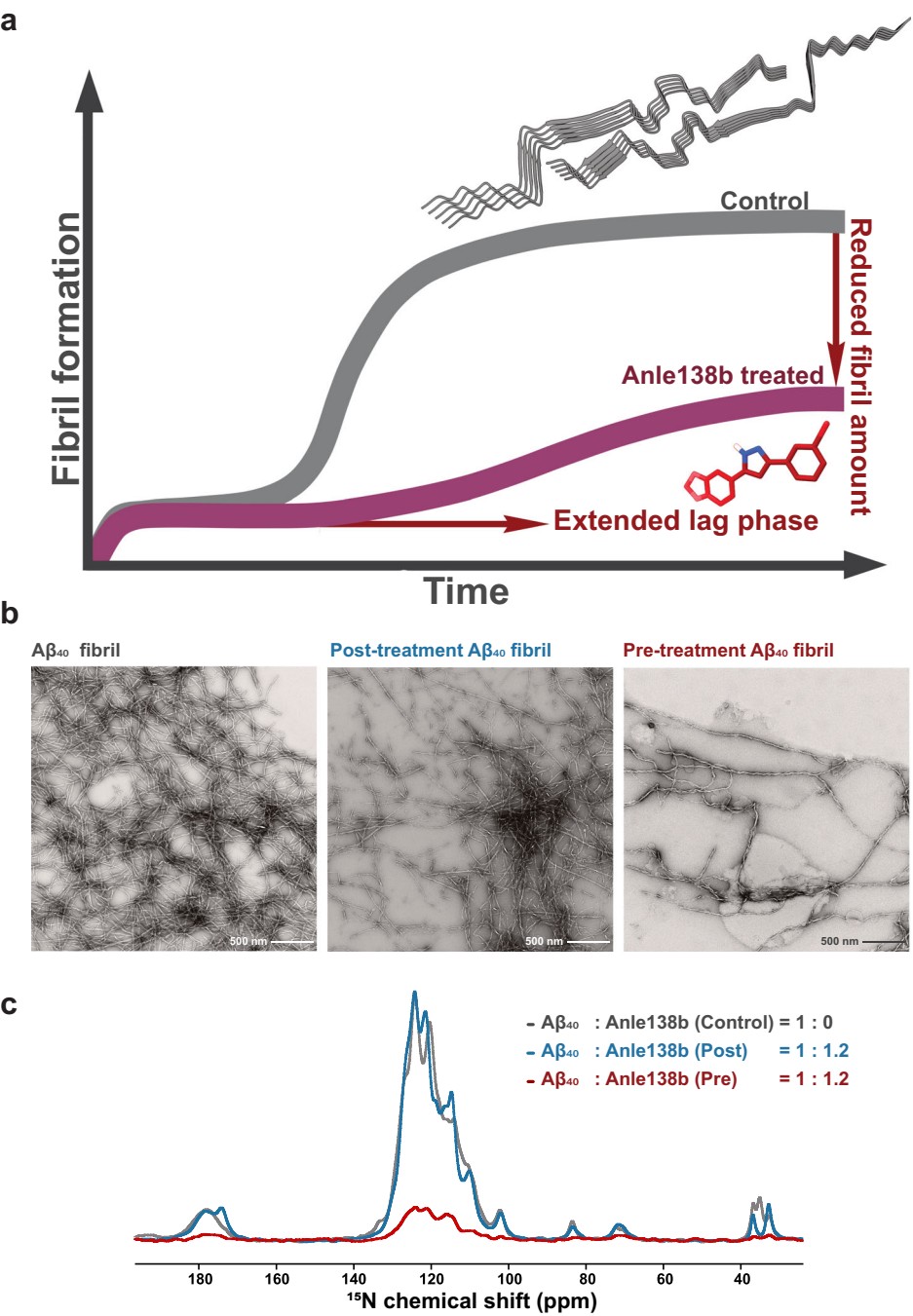

**Fig. 1 | Anle138b modulates amyloid fibril formation: Comparison between pre-treatment and control conditions. a** Schematic representation of ThT fluorescence data showing fibril formation in the presence (red, pre-treatment condition) or absence (gray) of anle138b. The presence of anle138b extends the lag phase and reduces the final amount of fibrils. Full experimental data are provided in Supplementary Fig. 1. **b** Negative stain electron microscopy (EM) images of Aβ40 fibrils. Left: Control fibrils formed without anle138b. Middle: Fibrils that were post-treated with anle138b (anle138b: Aβ40 molar ratio (SMPR) = 1.2). Right: Fibrils formed in the presence of anle138b (pre-treatment condition, SMPR = 1.2). **c** 1D ($^1$H)$^{15}$N CP solid-state NMR spectra of Aβ40 fibrils. Gray, blue, and red spectra correspond to control, post-treatment, and pre-treatment Aβ40 fibrils. Spectral intensity in the pre-treatment sample is ~25% of the control, consistent with ThT assay results and indicating reduced fibril formation in the presence of anle138b.

control-fibril sample, and the L1 structure was reconstructed from the cryo-EM data for each preparation (control fibril, post-treatment fibril, and pre-treatment fibril conditions). The cryo-EM analysis revealed no discernible structural differences between L1 Aβ40 fibrils in the presence or absence of anle138b, suggesting that the compound does not significantly alter the L1 Aβ40 fibril architecture (Fig. 2).

By contrast, ssNMR spectra revealed clear chemical shift perturbations (CSPs) in the presence of anle138b. In the 2D (H)NCA

spectrum, substantial CSPs were detected for residues in the loop region between the second and third β-strand (Ala21–Gly33) (Supplementary Fig. 2b, c) of the L1 filament fold (Supplementary Fig. 5a). These spectral differences, relative to the untreated (control) fibrils, were observed following anle138b post-treatment of L1 Aβ40 fibrils (SMPR 1.2). Notably, the large CSP values of up to 10 ppm in $^{15}$N and ~1.5 ppm in $^{13}$Cα suggest a direct interaction between anle138b and the fibril (Supplementary Fig. 5a).

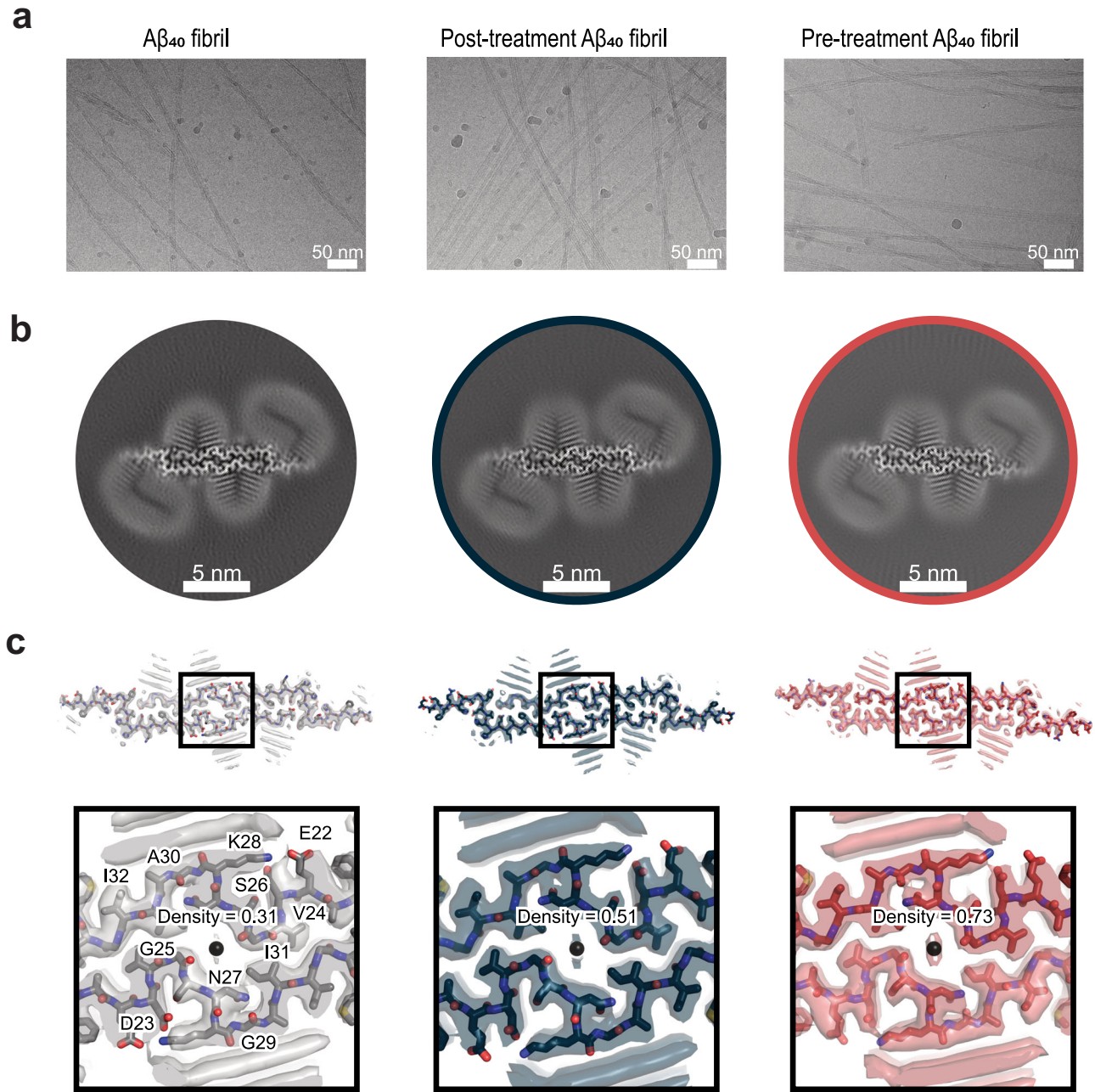

**Fig. 2 | Cryo-EM maps reveal additional non-proteinaceous densities in L1 Aβ₄₀ fibrils with anle138b.** Lipidic Aβ₄₀ fibril data in the absence of anle138b were previously published and is included here for comparison. **a** Exemplary 20 Å low-pass filtered cryo-EM micrographs and 2D class averages for different fibril pre-parations: control condition (without anle138b), post-treatment condition, and pre-treatment condition. **b** Cross-sections of the cryo-EM maps showing L1 Aβ₄₀ fibrils under the three conditions. **c** Overlay of a sharpened high-resolution cryo-EM map (transparent surface representation) and the atomic model (stick representation) of the fibril: control condition: gray (without anle138b), post-treatment condition: blue, and pre-treatment condition: red.

Similar localized CSPs were observed under the pre-treatment condition (SMPR 1.2). This observation suggests that anle138b interacts with the loop region for both pre- and post-treatment conditions (Supplementary Fig. 5). Since these CSPs are observed across different parts of the L1 Aβ₄₀ fibril structure, the data indicate that anle138b may bind to multiple regions of the fibril, including the central cavity of the two protofilaments and also to exposed fibril surfaces. However, CSPs can also be the result of allosteric effects (vide infra).

In conclusion, the CSP results indicate that anle138b likely binds in the loop region (Ala21–Gly33), which includes the central cavity formed at the interface between the symmetry-related loop regions of the two protofilaments within the L1 Aβ₄₀ fibril. Cryo-EM analysis

indicates that the L1 Aβ₄₀ fibril remains the predominant species in the presence of anle138b. The global fibril structure also remains unchanged.

## Anle138b binding in the central cavity of L1 Aβ₄₀ fibrils
To directly assess binding locations, we employed amino acid-specific isotope labeling strategies and DNP-enhanced NMR. Chemical shift perturbations (CSPs) may be due to direct binding of anle138b to specific amino acids of L1 Aβ₄₀ fibrils but could also be induced by allosteric structural changes. Therefore, DNP-enhanced NHHC experiments are required to localize the binding sites directly. For this purpose, the protein was labeled first uniformly with ¹³C and then with

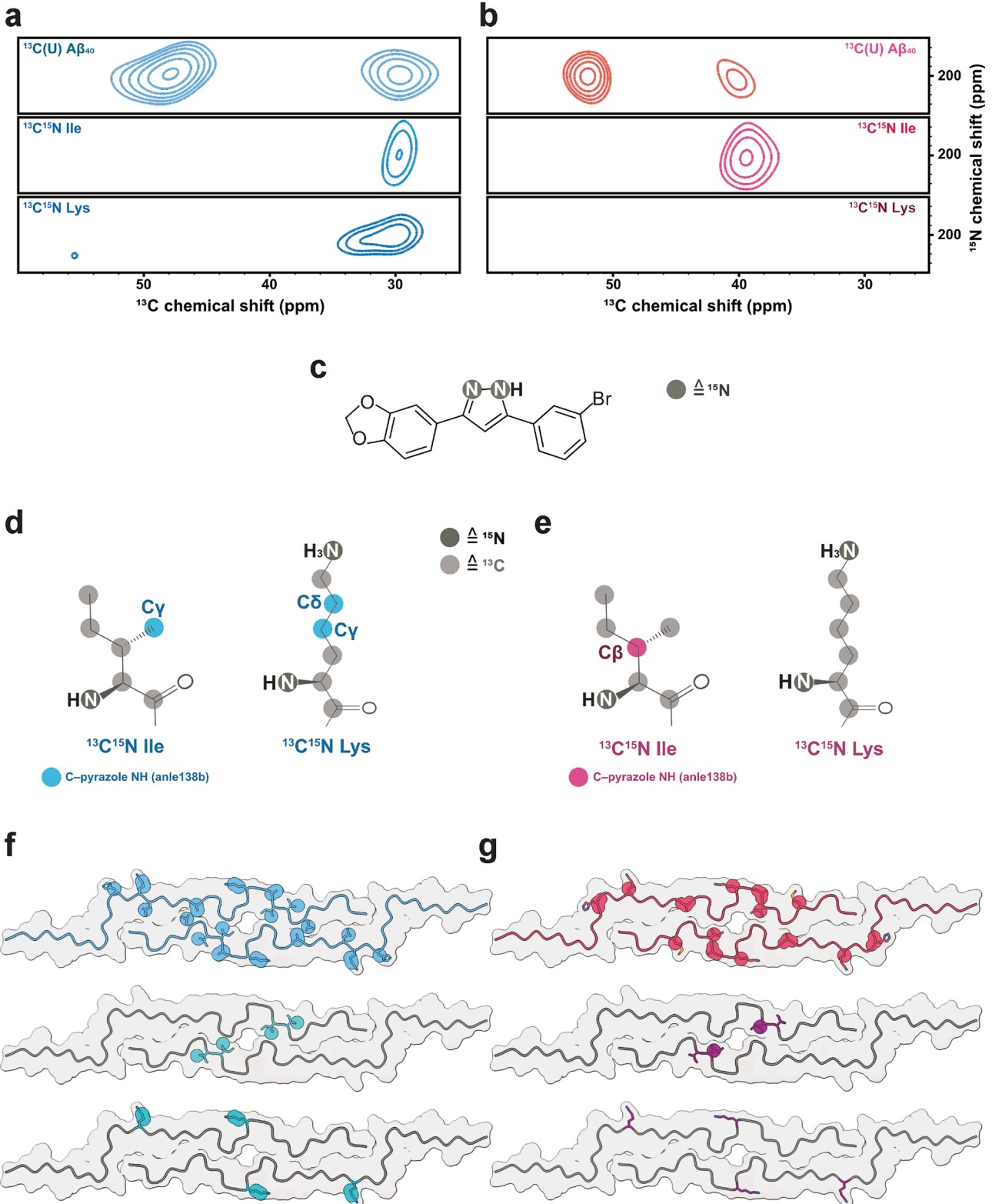

¹³C, ¹⁵N-isoleucine or ¹³C, ¹⁵N-lysine, while anle138b was uniformly labeled with ¹⁵N at the central pyrazole group. The cross-peaks in the NHHC spectrum[22] indicate close contact with the nuclei of the protein: the pyrazole ¹⁵N nuclei at 195 ppm are within approximately 5 Å of aliphatic ¹³C from the protein. These results provide direct evidence that anle138b binds to specific amino acids within the fibril (Fig. 3a, b), as detailed below.

Under the pre-treatment condition, a cross-peak appears between anle138b and the protein in the Cα region of the spectrum centered at 52 ppm, and a second peak is observed at about 40 ppm. The candidate residues that are consistent with these peaks include the Cα of His13, Ala21, and Ala30, and the sidechain resonances of Lys16, Lys28, Ile31, and Met35 (Fig. 3b, g). For a ¹³C, ¹⁵N-isoleucine–selectively labeled fibril, a cross-peak corresponding to Ile31 was identified as the

**Fig. 3 | Interaction of anle138b with L1 Aβ_{40} fibrils: pre-treatment and post-treatment conditions. a** 2D NHHC spectra of L1 Aβ_{40} fibrils obtained after the addition of [15]N-labeled anle138b under post-treatment conditions (anle138b: Aβ_{40} molar ratio (SMPR) = 1.2, mixing time = 200 μs). The top panel shows uniformly [13]C-labeled L1 Aβ_{40} fibrils, while the middle and bottom panels show spectra of fibrils selectively labeled with [13]C, [15]N-Ile31,32, and Lys16,28, respectively. Cross-peaks were observed between the [15]N-labeled pyrazole group of anle138b and specific [13]C nuclei within L1 Aβ_{40} fibrils. **b** Same experimental setup as in **a**, except that anle138b was added before fibril formation (pre-treatment condition, SMPR = 1.2, mixing time = 200 μs). **c** Molecular structure of [15]N-labeled anle138b, with labeling sites shown as dark gray circles. **d** Chemical structures of labeled Ile

and Lys residues under post-treatment conditions. [13]C atoms identified in proximity to the pyrazole NH of anle138b are indicated with light blue circles. **e** Same as in **d**, except for the pre-treatment condition. A proximate [13]C atom is marked with a pink circle. **f** Mapping of identified contact sites onto the L1 Aβ_{40} fibril structure under post-treatment conditions. Sky blue spheres indicate [13]C atoms from residues in the L1 Aβ_{40} fibril that are potential interaction sites for anle138b. Turquois spheres in the middle and bottom models represent [13]C atoms within Ile31,32 (middle, light turquoise) and Lys16,28 (bottom, dark turquoise) that are in direct contact with the pyrazole NH of anle138b. **g** Same as in **f**, but for pre-treatment conditions, with red and dark purple spheres indicating [13]C atoms in direct contact.

interacting residue based on the Cβ chemical shift (Fig. 3b, g). In contrast, no cross-peaks were observed for a [13]C, [15]N-lysine–selectively labeled fibril, thereby excluding Lys16 and Lys28. Given that Ile31 points towards the central cavity and no contacts were detected with the lysine sidechains on the fibril surface, these data suggest that anle138b predominantly interacts with residues in the central cavity under the pre-treatment condition. The 52 ppm peak is consistent with contact to residue Ala30, which is next to Ile31.

In contrast, under the post-treatment condition, two prominent peaks corresponding to anle138b are observed at different positions, namely 49 ppm and 30 ppm (Fig. 3a, f) in the 2D NHHC spectrum of [13]C-labeled protein. The Cα of residues such as Ala21, Ala30, Gly25, and Gly33 is consistent with the former peak, and the 30 ppm peak is consistent with the resonances of His14 (Cβ), Lys16, Lys28 (Cγ, Cδ), Ile31, Ile32 (Cγ1), Leu17, and Leu34 (Cδ). Leu17, Gly33, and Leu34 were excluded from consideration because of their buried location within the fibril. A DNP-enhanced 2D NHHC spectrum of isoleucine-labeled fibrils confirmed an interaction with Ile31 or Ile32. Lysine labeling also revealed a cross-peak at 30 ppm (Lys Cγ, Cδ), which, considering that these sidechains extend from the fibril, places anle138b on the fibril surface. These data confirm that anle138b interacts with surface-exposed residues such as Lys16, Lys28, and potentially Ile31 and/or Ile32. However, the small chemical shift difference (<1.35 ppm) between Ile31 Cγ1 and Ile32 Cγ1 resulted in peak overlap, preventing definitive assignment of the observed cross-peak to either residue. Therefore, binding to the central cavity cannot be definitively confirmed or excluded under the post-treatment condition.

Interestingly, the cryo-EM structures revealed an additional weak non-proteinaceous density within the fibril's central cavity, going from 0.31 without anle138b to 0.51 in the post-treatment fibril condition to 0.73 in the pre-treatment fibril condition (Fig. 2c, d; Supplementary Fig. 6). Non-proteinaceous density (primarily lipids)[9] is observed outside the fibrils, but there is no systematic density change depending on the presence of anle138b, unlike in the central cavity.

## Post-treatment titration of the anle138b-L1 interaction

To assess the binding behavior under post-treatment fibril conditions, we characterized concentration-dependent interactions using ITC and NMR.

Isothermal titration calorimetry (ITC) revealed that anle138b binds to L1 Aβ_{40} fibrils with an exothermic and spontaneous profile (ΔH = −1.84 kcal/mol, ΔG = −8.45 kcal/mol). The dissociation constant (K_d) was determined to be 0.64 μM, with a binding stoichiometry of ~0.72 molecules per Aβ_{40} monomer (Fig. 4b, Supplementary Table 2). CD spectroscopy and ThT fluorescence confirmed full conversion of Aβ_{40} monomers into fibrils under both control and post-treatment conditions, consistent with near-complete sedimentation of L1 Aβ_{40} after ultracentrifugation (Supplementary Fig. 3).

To further assess drug incorporation, we analyzed the supernatant after ultracentrifugation using 1D solution NMR. No free anle138b signal was detected at anle138b: Aβ_{40} ratios of 0.2:1 and 0.8:1, while ~30% of the drug remained unbound at 1.2:1 (Supplementary Figs. 21, 22). These results align with the stoichiometry measured by

ITC (Fig. 4b, Supplementary Table 2) and NMR titration (Fig. 4a). Together, they indicate that the majority of anle138b associates with the fibrils.

DLPG vesicles lacking anle138b produced no measurable heat signal in ITC (Supplementary Fig. 13b, Supplementary Table 2). This absence of detectable interaction was corroborated by ssNMR NOE and cryo-EM, which showed no DLPG-specific contacts or densities on the L1 Aβ_{40} fibril surface (Figs. 2b, 5).

To resolve residue-specific changes and explore whether the binding stoichiometry observed in ITC correlates with the NMR spectra, we performed an NMR-based titration experiment using SMPRs of 0.2, 0.4, 0.8, and 1.2. The effects of binding were monitored via 3D (H) CANH spectra, analyzing CSPs and signal intensity changes (I_ratio). This revealed large chemical shift changes and slow exchange behavior in several residues (Fig. 4a).

Peaks associated with anle138b-bound fibrils emerged as early as SMPR 0.2 across the β2 strand, loop, and β3 regions, with residue- and region-specific differences in saturation behavior. Several residues exhibited both free and bound peaks between SMPR 0.2 and 0.8, with most sites saturating at 0.4 or 0.8. Notably, Gly29, Ala30, and Asn27 required SMPR 1.2 to reach saturation (Fig. 4a, Supplementary Fig. 11–13).

In more detail, in the β2 region, Leu17 showed early binding at SMPR 0.2 and saturated at 0.4, whereas Lys16 remained unaffected until signal loss at SMPR 1.2. In the loop region, multiple residues responded at low concentrations: Ala21, Gly25, and Gly33 reacted at 0.2, with Ala21 saturating at 0.4. Other loop residues (Gly29, Ala30, Ile31) responded at 0.4, while Asn27 showed delayed binding (slow exchange at 0.8; saturation at 1.2). Notably, Lys28 and Asp23 showed concentration-dependent signal disappearance at 0.4 and 0.8, respectively, and Ser26 was undetectable at 1.2. In the β3 region, Met35 responded early (0.2), and Leu34 entered slow exchange at 0.4 and saturated at 0.8. Together, these observations highlight region-specific differences in binding sensitivity, with the loop region displaying the broadest range of response behaviors.

Significant [15]N chemical shift perturbations were observed for Ala30 (10.72 ppm), Ala21 (6.48 ppm), and Ile31 (4.65 ppm) in CSP-based titration experiments conducted under saturation conditions (SMPR = 1.2) (Fig. 4a). These large shift changes in amide nitrogen signals can result from changes in the hydrogen bonding environment, and do not necessarily reflect a structural change in the protein[23–27]. Indeed, upon anle138b binding, water-protein NOE contacts in the 3D H(H)NH NOE spectrum increased for Gly29, Ala30, Ile31, and Ile32, while decreasing for Gly25 and Leu34 (Supplementary Fig. 17).

The same spectrum also revealed that lipid interactions between β-strand residues (Leu17–Val8, Leu34–Val36) and DMPG acyl chains were disrupted, while new lipid contacts were formed with loop residues Ala21–Gly25 (Fig. 5). The disappearance of Asp23 and Lys28 signals indicates changes in the structure or dynamics of charged sidechains at the fibril surface near the loop region. Previous studies reported that Asp23 and Lys28 exist in two distinct populations (Asp23/Asp23′ and Lys28/Lys28′), of which only the Asp23–Lys28 pair forms a salt bridge[9]. In the 2D [13]C[13]C-DARR spectra acquired under the

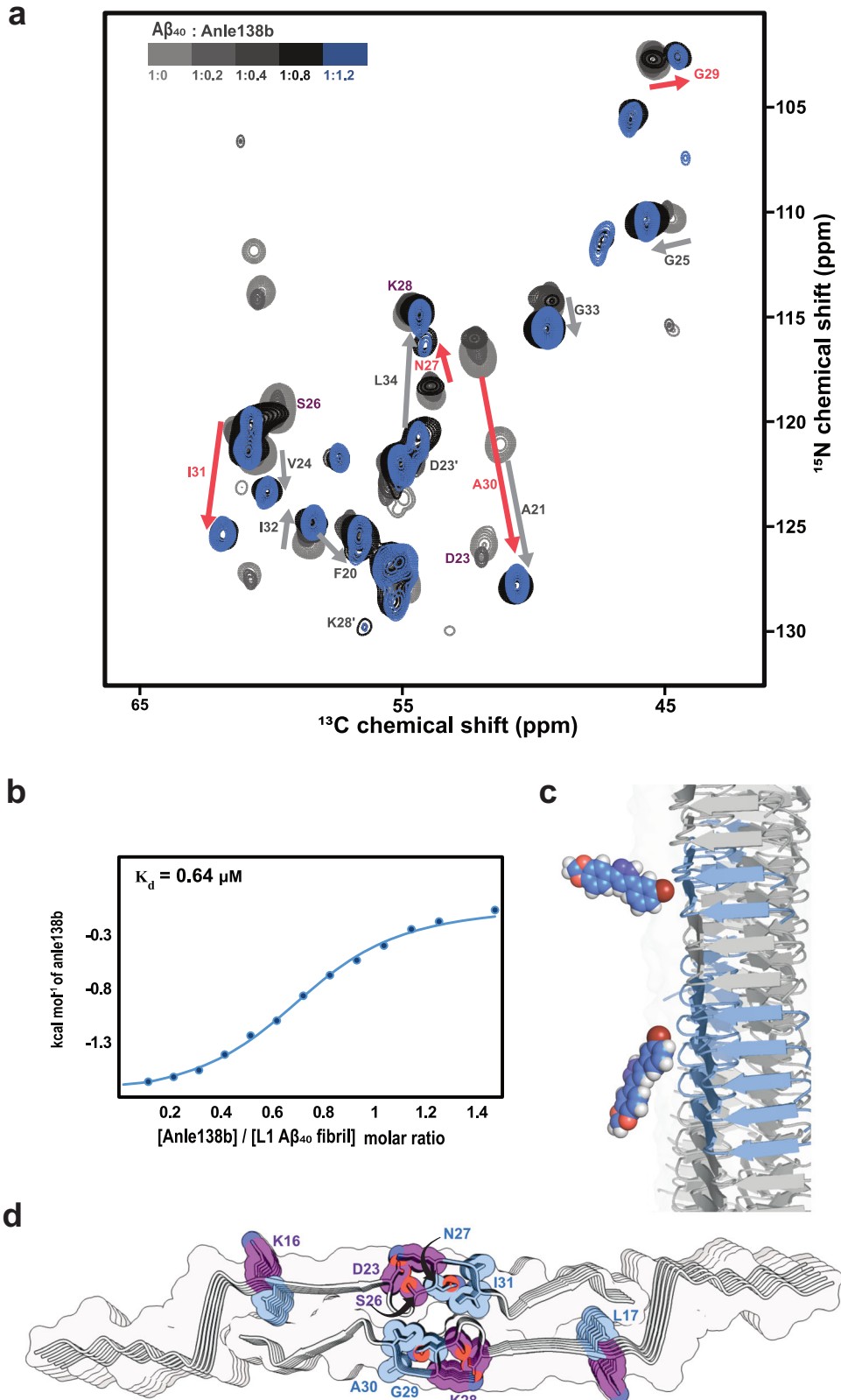

SMPR 1.2 condition (mixing times = 20 and 200 ms), the peaks corresponding to the salt bridge conformation were absent, while only those corresponding to Asp23′ and Lys28′ were observed (Fig. 4a, Supplementary Fig. 8a, 15, 16). In addition to these residues, Ser26 also exhibited peak loss, and no cross-peaks were detected among Asp23(Cβ), Ser26(Cβ), Asn27(Cα/Cβ), and Lys28(Cε). Given that CP-based DARR spectra selectively detect rigid regions of the protein,

these observations suggest enhanced flexibility or dynamic averaging in the loop region upon anle138b binding. At cryogenic temperature, differences in the chemical shifts of Lys28 Cγ and Cδ signals are evident: the room temperature assignments match pre-treatment fibrils, and these signals are absent in the post-treatment fibrils (Supplementary Fig. 10a). No major differences between pre- and post-treatment fibrils were observed for the isoleucine ¹³C resonances in the

**Fig. 4 | Titration of anle138b binding to L1 Aβ40 fibrils under post-treatment conditions. a** 3D (H)CANH solid-state NMR spectra of L1 Aβ40 fibrils titrated with increasing concentrations of anle138b (anle138b: Aβ40 molar ratio (SMPR) from 0 to 1.2). The spectrum shown is a 2D projection of the 3D spectrum. Chemical shift perturbations (CSPs) and signal intensity reductions are observed upon treatment with anle138b. Residues exhibiting both CSPs and signal attenuation are marked with pink arrows; those showing CSPs only are marked with gray arrows. Peaks for residues K16, D23, S26, and K28 (highlighted in purple) become undetectable at the highest anle138b concentration. **b** ITC binding isotherm for anle138b binding to L1 Aβ40 fibrils, derived from integrated injection heats (solid blue circles) and fitted with a least-squares fit (blue line). The fitted binding isotherm yields a dissociation constant ($K_d$) of 0.64 µM and a binding stoichiometry of -0.72 anle138b molecules per Aβ40 monomer. Source data are provided as a Source Data file. **c** Structural modeling suggests two illustrative binding orientations of anle138b on the fibril surface: one approximately perpendicular (-90°) and the other parallel (-0°/180°) to the fibril axis, bridging up to 2 or 5 β-sheet layers, respectively. **d** Perturbed residues are mapped onto the fibril structure. Blue indicates residues with both CSP ≥ 1.0 and intensity ratio ($I_{bound}/I_{free}$) ≤ 0.6. Purple indicates residues with undetectable signals at SMPR 1.2. The oxygen and nitrogen atoms are depicted in red and dark blue, respectively.

2D $^{13}C^{13}C$ -DARR spectra acquired at cryogenic temperatures (Supplementary Fig. 10a). Note that the linewidths of $^{13}C$ resonances at cryogenic temperature limit the detection of more subtle local differences. Taken together with cryo-EM analyses showing that all three fibril preparations—pre-, post-treatment, and control—share the same overall structure (Fig. 2), these findings suggest that the observed changes in the loop region primarily reflect enhanced local flexibility and water/lipid exposure rather than stable or static structural rearrangements.

## Surface and central cavity binding modes explored via MD simulations

All-atom molecular dynamics (MD) simulations were employed to investigate the binding of anle138b to the L1 Aβ40 fibril structure observed by cryo-EM and NMR. We first considered the binding of anle138b, in the presence or absence of DMPG lipids, corresponding to an L1 Aβ40 fibril structure with and without surface-bound phospholipids (see Methods, Fig. 6, Supplementary Fig. 23). In both simulation sets, anle138b molecules established contacts with regions of the fibril surface, consistent with the NMR results of post-treatment fibrils (Fig. 6a–c). Notably, anle138b did not bind to the central cavity in either lipid-containing or lipid-free simulations. Lipid-free simulations show short distances (below 5 Å) between the pyrazole NH of anle138b and Cδ of residue Ile32, consistent with NMR data, while the lipid-containing simulations do not, suggesting competition of lipid and anle138b binding at the L1 Aβ40 fibril surface near the position of Ile32 (Fig. 6b, c, Supplementary Fig. 18a, b).

In a second set of simulations, we probed the ability of anle138b to spontaneously bind to the central cavity between the two protofilaments of the L1 Aβ40 fibril as observed under both pre-treatment fibril and post-treatment fibril conditions. To do so, the diffusion of anle138b in the simulation box was restricted to a cylindrical volume centered on the loop region by a flat-bottomed position restraint (see Methods, Supplementary Fig. 19a, b). Indeed, anle138b can insert spontaneously in the central cavity of the loop region, showing contacts with Val24, Gly25, Ser26, Asn27, and Ile31 (Fig. 6a, b). During the microseconds-long MD trajectories, anle138b moved into the filament in eight out of ten simulation replicas, traversing up to 5 or 6 monomer layers into the central cavity. Once fully buried inside the fibril, the anle138b molecules are retained, as no unbinding events were observed (Supplementary Fig. 19a, b).

Interestingly, MD simulation models with deprotonated Lys28 and hence weakened side chain interactions between residues Asp23 and Lys28, allowed for a higher anle138b insertion rate (ten out of ten simulation replicas) and increased probability of deeper penetration into the central cavity of the fibril (Supplementary Fig. 19c). The disruption of the Asp23-Lys28 salt bridge led to an increase in RMSF values in the loop region, indicating enhanced structural flexibility (Supplementary Fig. 19d).

Anle138b molecules binding to the fibril surface and inside the central cavity exhibited contact patterns to residues Lys16, Lys28, Ile31, and Ile32, reproducing the two principal anle138b binding modes observed in the pre- and post-treatment fibril conditions by ssNMR experiments (Fig. 6a, c). Examination of the molecular determinants of anle138b binding revealed that the surface binding around the loop region occurred through contacts to the polar side chains of Asp23 and Lys28 (Fig. 6d, e). Specifically, the MD simulations revealed that the bromine (Br) atom of anle138b interacts with the ε-amino nitrogen (N) of Lys28 through a halogen bond-like contact[28] (Fig. 6d, Supplementary Fig. 20a), whereas the central cavity binding was dominated by close polar interactions between anle138b and the exposed protein backbone atoms, particularly of Gly25 (Fig. 6e). The simultaneous interactions via the pyrazole and bromophenyl moieties allow for a favorable binding of anle138b to the fibril.

Compared to the distribution in the bulk lipid phase, the bromophenyl ring of surface-bound anle138b orients preferentially such that the bromine and pyrazole nitrogen atoms align on the same edge of the molecule (Supplementary Fig. 20b). Interestingly, the prominent binding to Asp23 and Lys28 with this ring orientation leads to a perpendicular alignment of anle138b with respect to the fibril axis and allows for close stacking of the ligands (Supplementary Fig. 20c). In contrast, anle138b binding to other regions of the L1 Aβ40 fibril surface, such as Lys16 or Gly37, as well as the central cavity, occurred with the molecules aligned parallel to the fibril axis.

In contrast to surface binding, the cavity binding mode showed no preferred bromophenyl ring orientation or anle138b alignment (bromophenyl moiety pointing down (0°) or up (180°) with respect to the fibril axis, Supplementary Fig. 18c, 20c, d). The observed anle138b binding mode characteristics were independent of the presence of DMPG lipids in the simulations, as well as the anle138b tautomer (Supplementary Fig. 20c, d).

## Discussion
### Disease relevance of the action of anle138b on L1 Aβ40 fibrils
This study builds on previous findings that L1 Aβ40 fibrils share structural similarity with Aβ40 fibrils amplified from amyloid seeds derived from the brain tissue of Alzheimer's disease (AD) patients (PDB-ID: 6W0O)[10]. Similar fibril architectures have also been reported in Aβ40 fibrils amplified from vascular tissue from cerebral amyloid angiopathy (CAA) patients, including the familial Dutch variant (fCAA-Dutch, PDB-ID: 8FF3)[11]. These structural parallels and their pathological relevance support the use of L1 Aβ40 fibrils as a suitable model system for studying amyloid aggregation associated with AD and CAA.

Anle138b effectively inhibited Aβ40 fibril formation under pre-treatment conditions, consistent with previous findings of reduced Aβ42 deposition in the APP/PS1 mouse model[17]. Under these conditions, anle138b predominantly binds within a central cavity of the Aβ40 fibril, as revealed by ssNMR-detected contacts between anle138b and specific amino acids and corroborated by cryoEM density increase in the central cavity. This binding mode closely resembles that observed for α-synuclein[29,30], where internal cavities arise from the repetitive fibril architecture and imperfect core packing. Similar cavities have been reported in some polymorphs of other amyloid fibrils[31] and have been proposed to serve as favorable binding sites for small hydrophobic molecules[32,33]. In both L1 Aβ40[9] and α-synuclein[34] fibrils, such cavities are lined with glycine and isoleucine residues—glycine, due to its small size, may promote cavity formation, whereas isoleucine likely contributes to hydrophobic stabilization of the ligand.

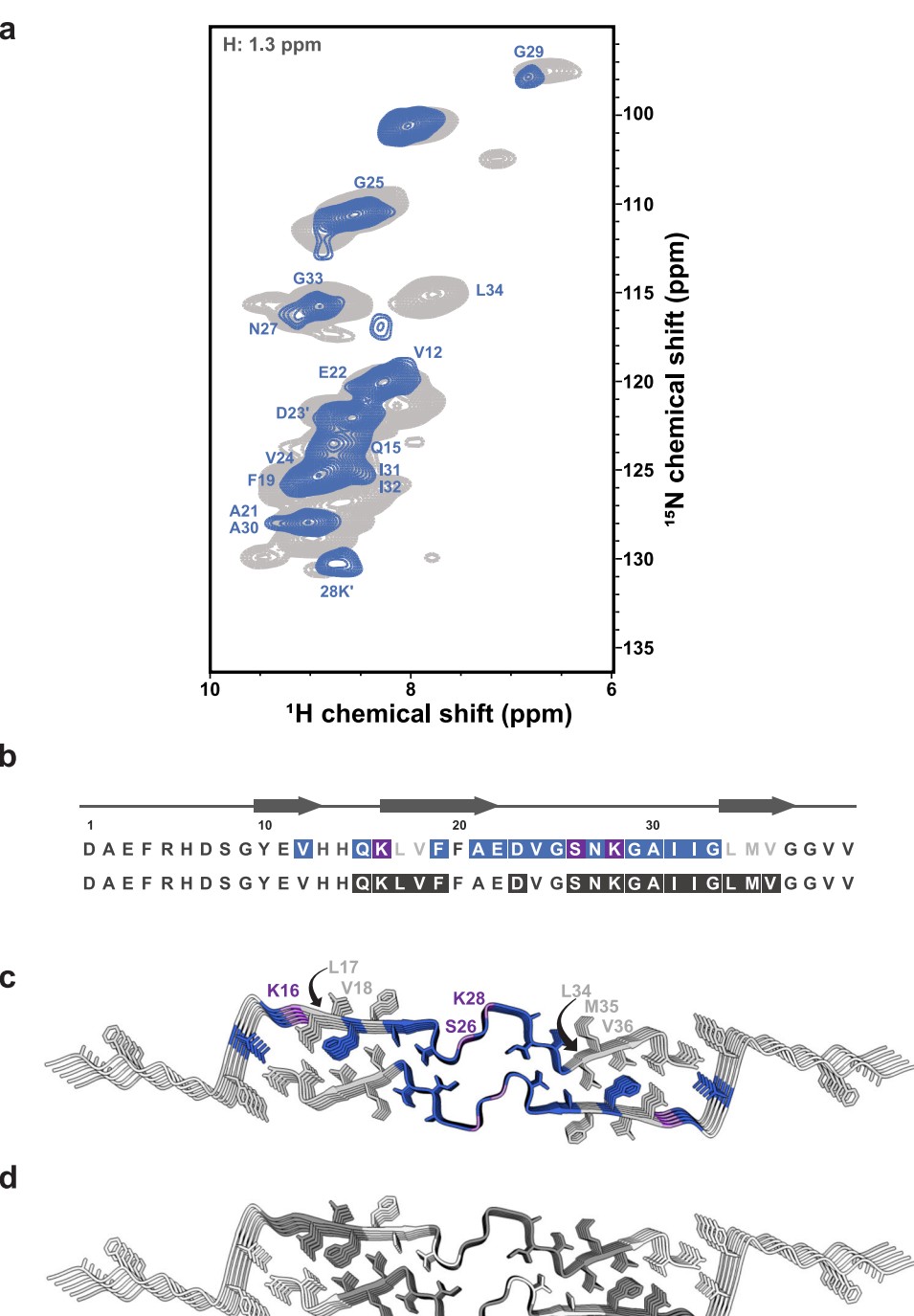

**Fig. 5 | 3D H(H)NH NOE analysis of lipid–fibril interactions in L1 Aβ₄₀ fibrils after anle138b treatment (post-treatment condition). a** 2D (H)NH spectrum (gray) and 2D NH plane extracted at 1.3 ppm in the H dimension (lipid CH₂ region) from a 3D H(H)NH NOE spectrum (blue, mixing time = 50 ms), acquired from ²H, ¹⁵N, ¹³C-labeled L1 Aβ₄₀ fibrils treated with anle138b (post-treatment; anle138b: Aβ₄₀ molar ratio (SMPR) = 1.2). Blue cross-peaks mark backbone amide groups in proximity with lipid acyl chains. **b** Amino acid sequence and secondary structure of L1 Aβ₄₀ fibrils. Blue boxes (top) indicate residues interacting with lipid acyl chains (post-treatment condition, SMPR = 1.2). Light gray letters indicate loss of lipid contact post-treatment. Dark gray boxes (bottom) indicate lipid-interacting residues in the absence of anle138b. **c** Structural mapping of lipid-contacting residues under post-treatment conditions. Blue color indicates interaction sites. Notably, hydrophobic residues (L17, V18, L34, M35, V36) are not involved in lipid interactions in the presence of anle138b. Purple labels mark residues that show signal loss after treatment. Hydrophobic side chains are shown as white sticks for reference (also in **d**). **d** The same fibril model highlights lipid-interacting residues in the absence of anle138b (dark gray).

In contrast, under post-treatment conditions, anle138b binds to the fibril surface without disrupting the overall fibril architecture, while still inducing localized structural perturbations. This binding mode resembles the interaction patterns of known amyloid-binding compounds such as Thioflavin-T, Congo Red[35,36], and various PET tracers[37–40], although high stoichiometric excess is used in most of these studies. NHHC spectra provided direct evidence of anle138b binding to surface-exposed side chains[40], indicating preferential interaction at solvent-accessible regions of the fibril. 3D (H)CANH[41] titration data showed region-specific binding dynamics at room

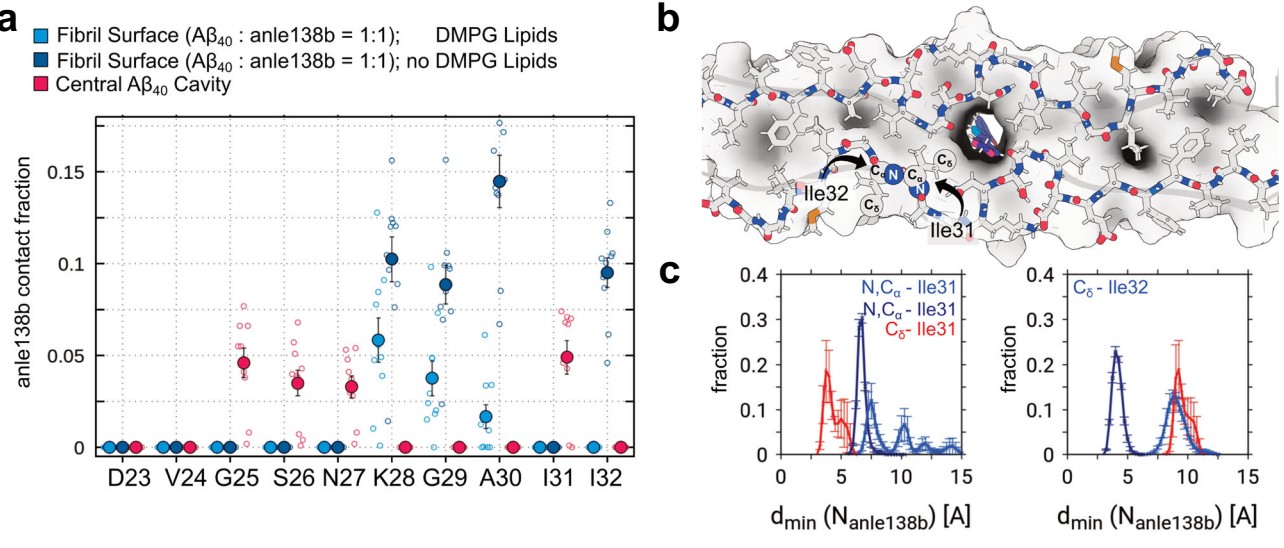

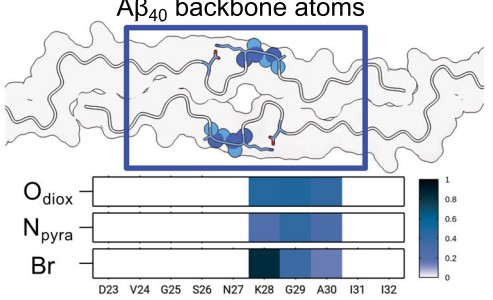

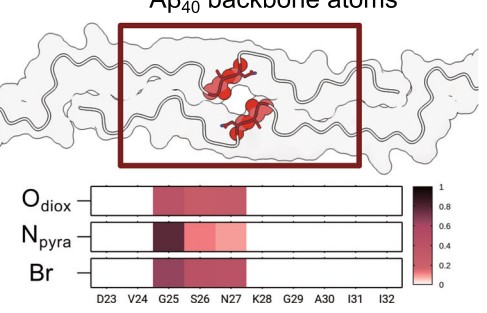

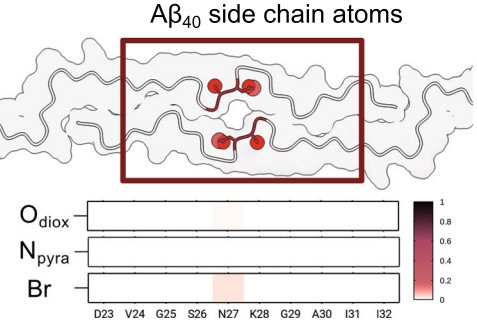

**Fig. 6 | Structural determinants and contact sites of anle138b bound on the fibril surface and to the central cavity of the loop region. a** Averaged anle138b contact fraction to the L1 $A\beta_{40}$ fibril structure for MD simulations with anle138b molecules either bound to the surface (with DMPG lipids, blue and without DMPG lipids, dark blue) or internally bound to the loop region (red). Contacts are considered if protein N or C atoms to anle138b pyrazole nitrogen atoms are found within a cutoff distance of 0.5 nm. Data are presented as mean ± s.e.m. (indicated by error bars) for 10 replicates, respectively. **b** Representative MD simulation snapshot of L1 $A\beta_{40}$ fibril loop region with internally bound anle138b viewed up close and down the long axis of the fibril. Black arrows indicate I31 and I32. **c** Distributions of distances between those N and/or C atoms of residues I31 and I32 that are minimal

to anle138b pyrazole nitrogen atoms for the simulation sets representing either surface-bound (blue and dark blue) or internally bound (red) anle138b. Closest atomic contact sites of anle138b pyrazole nitrogen atoms are highlighted by spheres in **b** (annotated only for one of the protofilaments for simplicity). Data are presented as mean ± s.e.m. (indicated by error bars) for 10 replicates, respectively. Renderings of L1 $A\beta_{40}$ fibril loop region detailing contact sites accessible either to **d** Surface-bound or **e** Internally bound anle138b. Atoms from the backbone (left) and side chain (right) of the residues are highlighted separately. Renderings are accompanied by heat maps that report polar contacts between individual $A\beta_{40}$ residues and polar moieties of anle138b, respectively. Scale bars indicate contact probabilities. Source data are provided as a Source Data file.

temperature: in both β2 and β3 regions, some residues responded rapidly, while others exhibited slower exchange behavior. The loop region displayed a particularly distinct pattern—surface-facing sites saturated quickly, whereas inward-facing positions responded more gradually or only at higher ligand concentrations. Isothermal titration calorimetry (ITC) thermodynamically characterized the surface binding of anle138b to L1 $A\beta_{40}$ fibrils, revealing exothermic binding with a

dissociation constant ($K_d$) of 0.64 µM and a stoichiometry of ~0.72 molecules per monomer. This result was consistent with NMR titration data, which similarly indicated a stoichiometry of 0.7–0.8 molecules per monomer and revealed residue-specific differences in saturation behavior, suggesting that anle138b binds selectively to defined structural regions. These results are further supported by ultracentrifugation analysis of the supernatant under post-treatment conditions,

confirming near complete conversion of Aβ$_{40}$ into fibrils and a substantial incorporation of anle138b into the fibril fraction.

The observed substoichiometric binding ratio can be explained with multiple binding sites for anle138b in surface-exposed regions of the fibril, rather than the central cavity. Structural modeling suggests that anle138b does not bind individually to each monomer but instead interacts with interfacial β-sheet surfaces that span multiple stacked monomer layers. When oriented parallel to the fibril axis, a single anle138b molecule may span up to five β-sheet layers or up to two layers when aligned perpendicular, providing a structural rationale for the ITC-derived binding stoichiometry.

The loop region of L1 Aβ$_{40}$ fibrils (Ala21–Gly33) plays a critical role in forming the central cavity between the two protofilaments. DNP NHHC spectra reveal that, under post-treatment with anle138b, the compound binds to Cγ atoms of Ile and the Cγ and Cδ atoms of Lys28. These residues are located within the loop region and exposed on the fibril surface, indicating that anle138b interacts with the fibril at the surface under these conditions. Cryo-EM analysis confirms that the overall fibril architecture remains intact. However, the attenuation of loop residue signals in room-temperature CP spectra suggests that anle138b binding enhances the dynamic motion of the loop rather than inducing static structural changes. In particular, residues such as Ala21, Ala30, and Ile31 showed pronounced CSPs and concomitant changes in hydration, indicating that ligand binding perturbs their local physicochemical environment within the loop region. Furthermore, Gly25, located within the central cavity, shows a loss of hydration, consistent with displacement of water from the cavity upon anle138b binding. This interpretation is supported by the loss of the Asp23–Lys28 salt bridge on the loop surface, suggesting that anle138b binding correlates with increased loop flexibility. While solid-state NMR data indicate interactions at the fibril surface in the post-treatment fibrils, molecular dynamics simulations suggest an additional or complementary binding mode: upon deprotonation of Lys28, loop flexibility increases further, facilitating anle138b insertion into the central cavity.

Our results reveal a dual mode of action: inhibition of fibril formation during early aggregation and selective binding to mature fibrils without major structural disruption. From a thermodynamic perspective, it is intriguing that anle138b can stably associate with fibrils while simultaneously impeding their formation. Although the structural basis for the observed inhibition remains unresolved, one plausible explanation is that anle138b preferentially stabilizes early, non-fibrillar aggregates, thereby reducing the formation of mature fibrils. This mechanism is similar to that proposed for anle145c, a structurally related diphenyl-pyrazole (DPP) compound, which has been shown to inhibit hIAPP fibrillation and has been proposed to stabilize non-toxic oligomeric species through a thermodynamically driven process[42]. Moving forward, we aim to identify and structurally characterize these compound-stabilized early intermediates, which may hold the key to understanding the therapeutic mechanism of anle138b.

## Methods

### Protein expression and purification

Aβ$_{40}$ fused to an N-terminal SUMO domain with a histidine tag was produced recombinantly, as previously described[9]. Briefly, uniformly $^{13}$C- and $^{13}$C,$^{15}$N-labeled samples were expressed in a minimal medium supplemented with $^{15}$NH$_4$Cl and $^{13}$C$_6$-D-glucose. $^2$H,$^{13}$C,$^{15}$N-labeled Aβ$_{40}$ was expressed in *E. coli* strain BL21(DE3) adapted to 100% $^2$H$_2$O minimal medium supplemented with $^2$H$_7$,$^{13}$C$_6$-β-D-glucose and $^{15}$N-ammonium chloride. For the production of amino acid-specific forward labeled protein ($^{13}$C,$^{15}$N-Lys, $^{13}$C,$^{15}$N-Ile), the labeled amino acids were added to the minimal medium. The fusion protein was purified on nickel NTA resin (Macherey-Nagel), followed by digestion with tobacco etch virus (TEV) protease. Subsequently, the digestion mix was supplemented with 6 M guanidinium hydrochloride, and the SUMO

domain and TEV protease were removed by another passage over the nickel resin. Final protein purification was performed by reversed phase HPLC on a C4 Vydac column. The purified peptide was finally lyophilized before use.

### Preparation of [1,2-$^{15}$N$_2$]−3-(1,3-benzodioxol-5-yl)−5-(3-bromophenyl)−1H-pyrazole

The synthesis of isotope-labeled anle138b derivatives was carried out as described[29]. Briefly, [1,2-$^{15}$N$_2$]-anle138b was obtained by condensation of [$^{15}$N$_2$]hydrazine sulfate with the corresponding diketone[12], and [3,4-$^{13}$C$_2$]-anle138b was synthesized via triflic acid-mediated Friedel–Crafts acylation of 1,3-benzodioxole with [1,2-$^{13}$C$_2$]acetic acid followed by Claisen condensation and Knorr cyclization. The products were confirmed by NMR and mass spectrometry, and detailed procedures have been reported earlier[29].

### L1 Aβ$_{40}$ fibril preparation

The L1 Aβ$_{40}$ fibril was prepared as described in a previous report[18]. Briefly, lipid films were generated from either 1,2-dimyristoyl-sn-glycero-3-phosphoglycerol (DMPG) or 1,2-dilauroyl-sn-glycero-3-phosphoglycerol (DLPG) by evaporating the solvent under a stream of N$_2$, followed by lyophilization. Unless otherwise noted, all experiments were conducted in 10 mM phosphate buffer, pH 6.5, at 37 °C.

For fibril formation experiments, the DMPG film was hydrated and sonicated in 10 mM sodium phosphate buffer (pH 6.5) for 5 minutes to generate vesicles that promote Aβ$_{40}$ aggregation. Under the pretreatment fibril condition, anle138b was incorporated into DMPG during lipid film preparation to produce vesicles containing both lipid and anle138b (LPR = 30:1, SMPR = 0.2-1.2). Control fibrils and pretreatment fibrils were formed with DMPG liposomes with or without anle138b.

The Aβ$_{40}$ stock solution was diluted into buffer containing either DMPG liposomes or anle138b-loaded DMPG liposomes (small unilamellar vesicles, SUVs; diameter 30–100 nm) to yield final concentrations of 20 μM protein and 600 μM lipid. Samples were incubated at 37°C for 1–2 days under quiescent conditions. Fibril formation was confirmed by circular dichroism (CD) spectroscopy and thioflavin-T (ThT) fluorescence.

For post-treatment fibril, DLPG lipid films containing anle138b were hydrated and sonicated for 15 minutes in 10 mM sodium phosphate buffer (pH 6.5), yielding lipid-based particles <10 nm in diameter to solubilize the hydrophobic compound. Although the exact structural identity of DLPG particles remains unclear, with their small size (< 10 nm) and physicochemical properties suggesting they are more likely micelles, non-bilayer structures, or ultrasmall unilamellar vesicles[43], we refer to them as vesicles throughout this work for clarity. For post-treatment conditions, DLPG vesicles were used to deliver anle138b to pre-formed L1 Aβ$_{40}$ fibrils. DLPG was selected over DMPG due to its lower melting temperature (T$_m$ = −3 °C), which ensures high membrane fluidity at 25°C and facilitates dispersion of hydrophobic molecules such as anle138b. To minimize cosedimentation artifacts during centrifugation, small unilamellar DLPG vesicles were used. Control ITC experiments confirmed that DLPG vesicles alone exhibited no measurable binding enthalpy with L1 Aβ$_{40}$ fibrils, supporting their role as a passive solubilizing vehicle for anle138b.

The anle138b-loaded DLPG vesicles were used to treat preformed L1 Aβ$_{40}$ fibrils for subsequent ITC, DNP-NHHC, HCANH titration, ssNMR, and Cryo-EM experiments. With the exception of the ITC experiment, L1 Aβ$_{40}$ fibrils were incubated at 37°C for 1 h to allow sufficient interaction prior to measurement. The fibrils were collected by high-speed centrifugation at 55,000 g for 60 minutes for subsequent ssNMR measurements. The pelleted fibrils were then packed into MAS rotors using a funnel consisting of a trimmed pipette tip and a tabletop centrifuge.

## Aggregation assays

Aggregation assays were conducted under the same lipidic conditions used for fibril formation, either with DMPG liposomes (SUV) alone (control fibril condition) or with DMPG liposomes (SUV) containing anle138b (LPR = 30:1, SMPR = 0.6 or 1.2).

$A\beta_{40}$ (20 μM) with liposome and thioflavin T (30 μM) were incubated in a non-binding black 96-well plate (Greiner Bio-One, item no. 655906) at 37 °C under quiescent conditions. Fluorescence signals were measured at 430 nm excitation (35 nm bandwidth) and 485 nm emission (20 nm bandwidth) using a Tecan Spark plate reader with double orbital shaking (1 min, 6 mm amplitude, 54 rpm, every 10 min). The manual gain was set to 40, with 30 flashes and an integration time of 40 μs. The z-position was calibrated using an empty well before each experiment. All measurements were recorded using Tecan Spark control software (v 2.2), and data were analyzed using Microsoft Excel.

## Solid-state NMR measurements

L1 $A\beta_{40}$ fibrils were prepared using the same protocol with various isotopic labeling schemes, including uniformly $^2H$, $^{13}C$, $^{15}N$-labeled, $^{13}C$, $^{15}N$-labeled, $^{13}C$-labeled, and selectively $^{13}C$, $^{15}N$-labeled (at Ile or Lys residues). These fibrils were used for backbone assignments, lipid interaction studies, intermolecular contact analysis, and investigations involving anle138b. Backbone assignments of lipidic $A\beta_{40}$ fibrils were not newly generated in this study but were transferred from previously published data[9]; BMRB 52006. To minimize errors in chemical shift transfer, buffer conditions and experimental parameters were matched to those reported previously. The transferred assignments were used for subsequent analysis of chemical shift perturbations and interaction studies.

The spectral similarity among the various L1 $A\beta_{40}$ fibril preparations was confirmed using multiple techniques and isotopic labeling schemes, including 2D (H)NCA and 2D $^{13}C-^{13}C$ DARR of $^{13}C^{15}N$-labeled fibrils, 2D $^{13}C-^{13}C$ DARR of $^{13}C$-labeled fibrils, 2D $^{13}C-^{13}C$ DARR of selectively labeled fibrils, and 3D (H)CANH of $^2H$, $^{13}C$, $^{15}N$-labeled fibrils. For residue-specific backbone assignments, 2D (H)NH, 3D (H)CANH, and 3D (H)coCAcoNH experiments were performed[41]. The 3D (H)CANH spectra were also acquired under different concentrations of anle138b for titration experiments. Carbon-detected spectra were acquired with the standard Bruker implementations found in the pulse program library, and proton-detected spectra according to the published pulse programs.[41,44] Detailed information about the parameters of each spectrum is provided in Supplementary Table 1.

All 2D $^{13}C-^{13}C$ DARR and 2D (H)NCA spectra were acquired on Bruker Avance III and NEO 850 MHz spectrometers equipped with 3.2 mm MAS HCN probes, operated at a magnetic field strength of 20.0 T, with MAS at 17 kHz and a sample temperature of 265 K. Titration experiments and 3D H(H)NH spectra (NOE mixing time = 50 ms) were recorded on $^2H$, $^{13}C$, $^{15}N$-labeled $A\beta_{40}$ fibrils using a 1.3 mm MAS HCN probe spinning at 55 kHz at 235 K, on a Bruker Avance III HD 800 MHz spectrometer operating at a magnetic field of 18.8 T[44]. A 3D (H) CANH spectrum of post-treatment Aβ40 fibrils (SMPR = 1.2) was acquired at 235 K on a Bruker Avance NEO 1200 MHz spectrometer (28.2 T) using a 1.3 mm MAS HCN probe at a MAS rate of 55 kHz. The hCANH spectrum acquired at 800 MHz validated the absence of spectral changes in the 1200 MHz spectrum.

Spectra were processed and analyzed using CcpNmr v2.5.2[45], and peak intensities were determined by integration of cross peaks. Assignments to lipidic $A\beta_{40}$ fibrils without anle138b have been reported previously[9]. Chemical shift perturbations were calculated as the average of the HN, NH, and Cα chemical shifts.

## DNP-enhanced ssNMR

TEMTriPol-1 was synthesized following established protocols[46]. The sample matrix, consisting of 45% $^{13}C$-depleted $d_8$-glycerol and 4 mM

TEMTriPol-1, was added to the fibril–lipid mixture and mixed to ensure homogeneous distribution.

All DNP experiments were performed on a Bruker Avance III HD 600 MHz spectrometer operating at 14.1 T, equipped with a 3.2 mm low-temperature (LT) MAS HCN probe. MAS frequencies used for each experiment are listed in Supplementary Table 1. Microwave irradiation at 395 GHz was generated using a gyrotron oscillator and transmitted via a corrugated waveguide. The sample temperature was maintained at ~100 K using a Bruker second-generation liquid nitrogen cooling system.

2D NHHC experiments (mixing time = 200 μs) were conducted on $^{13}C$-labeled and selectively labeled lipidic $A\beta_{40}$ fibrils in the presence of anle138b under both pre- and post-treatment fibril conditions. Additional 2D $^{13}C-^{13}C$ DARR and $^{13}C-^{13}C$ RFDR spectra were acquired on the same types of samples with anle138b applied in both conditions.

3D structural models were visualized using UCSF ChimeraX (v1.9)[47], and all graphs and plots were generated using Microsoft Excel.

## Affinity measurements

Isothermal titration calorimetry (ITC) was used to assess the binding affinity of anle138b to L1 $A\beta_{40}$ fibrils in 10 mM sodium phosphate buffer (pH 6.5) at 25°C, using a MicroCal PEAQ-ITC automated system (Malvern).

The experimental approach was based on established protocols for nanoparticle–protein interaction studies[48,49] and adapted to systems involving drug-loaded vesicles[50,51].

Anle138b (100 μM), prepared in DLPG vesicle (2 mM DLPG, particle diameter < 10 nm), was titrated into the sample cell containing L1 $A\beta_{40}$ fibrils (10 μM). The titration consisted of an initial 0.4 μL injection, followed by fourteen 2 μL injections, each lasting 4 seconds with 150-second intervals. As a control for dilution and background heat, DLPG vesicles without anle138b were titrated into the fibril solution using the same protocol. Control heat signals were subtracted from the raw thermogram to isolate the enthalpic contributions of anle138b binding. The ITC data were analysed using MicroCal PEAQ-ITC analysis software.

## Statistics and reproducibility

For each condition (control, pre-treatment, post-treatment), five isotopic labeling schemes were employed ($^1H$, $^{13}C$, $^{15}N$-labeled; $^2H$, $^{13}C$, $^{15}N$-labeled; $^{13}C$-labeled; Lys-selectively $^{13}C$, $^{15}N$-labeled; Ile-selectively $^{13}C$, $^{15}N$-labeled). NMR spectra were acquired for all samples and confirmed to display similar spectral patterns. Negative-stain EM was performed once per condition, confirming that fibrils longer than 1 μm, with comparable length and distribution, were formed across all samples. In addition, ThT fluorescence, CD, and ssNMR spectra were used to further confirm that the overall amounts of fibrils formed were comparable under all conditions. Following this validation, cryo-EM analysis was performed once each for the pre-treatment and post-treatment fibrils.

## Transmission electron microscopy

For negative staining, samples were applied to 400 mesh copper grids coated with glow-discharged carbon film. After staining with NanoVan (Nanoprobes Inc.), micrographs were taken at room temperature using a Talos L120C transmission electron microscope (Thermo Fisher Scientific).

## Cryo-EM sample preparation

When fibrils were formed at the same concentration of $A\beta_{40}$ monomer for cryo-EM analysis, the post-treatment condition produced a similar amount of fibrils as the control condition, whereas the pre-treatment condition produced relatively fewer fibrils. To normalize fibril concentrations, post-treatment fibrils were concentrated to 60 μM (as in previous studies)[9], and pre-treatment fibrils were concentrated to 720 μM, based on the initial $A\beta_{40}$ monomer

concentration. For this, 30 kDa Amicon concentrators (Ultra-500 µL) were used with a tabletop centrifuge at 12,000 g. After confirming comparable fibril amounts by CD spectroscopy, concentrated samples were used to prepare cryo-EM grids.

## Cryo-EM grid preparation and imaging

The cryo-EM procedures, including grid preparation, data acquisition, and motion correction, were largely performed as described in our previous study.[9] The key parameters that differed in this study are detailed below.

Grids were flash-frozen in liquid ethane using a Mark IV Vitrobot (Thermo Fisher) operated at 95% relative humidity. Data acquisition was carried out in EFTEM mode using a Quantum LS (Gatan) energy filter with a slit width of 20 eV. A total of 7311 and 21,576 movies were collected using SerialEM[52] for pre- and post-treatment condition samples, respectively. Each movie was recorded over 40 frames with a total accumulated dose of ~40.5 $e^-/Å^2$. The defocus range was set from −0.7 to −2.4 µm. Motion correction and dose-weighting were performed on-the-fly using Warp.[53]

## Helical reconstruction of Aβ40 fibrils

Aβ40 fibrils in complex with anle138b were reconstructed using RELION-3.1,[54] following the helical reconstruction scheme.[55] First, the estimation of contrast transfer function parameters for each motion-corrected micrograph was performed using CTFFIND4.[56] Next, filament picking was done using crYOLO.[57]

The reconstruction procedure was adapted from the original L1 Aβ40 fibril. Hence, for 2D classification, we extracted particle segments using a box size of 600 pix (1.05 Å/pix) downscaled to 200 pix (3.15 Å/pix) and an inter-box distance of 13 pix. For 3D classification, the classified segments post-2D classification were (re-)extracted using a box size of 250 pix (1.05 Å/pix) and without downscaling. Starting from a featureless cylinder filtered to 60 Å, several rounds of refinements were performed while progressively increasing the reference model's resolution. The helical rise was initially set to 4.75 Å, and the twist was estimated from the micrographs. Once the β-strands were separated along the helical axis, we optimized the helical parameters (final parameters are reported in Tab. S4). We then performed a gold-standard 3D auto-refinement, followed by standard RELION post-processing with a soft-edged solvent mask that includes the central 10 % of the box height yielded post-processed maps (B-factors are reported in Tab. S3). The resolution was estimated from the value of the FSC curve for two independently refined half-maps at 0.143 (Fig. S7). The optimized helical geometry was then applied to the post-processed maps yielding the final maps used for model building.

## Atomic model building and refinement

For the Aβ40 fibril, one protein chain was extracted from PDB-ID 8ovk. Subsequent refinement in real space was conducted using PHENIX[58] and Coot[59] in an iterative manner. The resulting models were validated with MolProbity,[60] and details about the atomic models are described in Table S4.

## MD simulation: protocol

The GROMACS 2023 simulation software package[61,62] was used to set up and carry out the MD simulations. Settings for production runs were chosen as follows: The long-range electrostatic interactions were treated using the Particle Mesh Ewald (PME) method.[63,64] Bonds in protein and lipid molecules were constrained using the P-LINCS[65] algorithm. Water molecules were constrained using SETTLE[66] algorithm. Neighbor lists were updated with the Verlet list scheme.[62,67] For production runs, the simulated systems were kept at a temperature of 300 K by applying the velocity-rescaling[68] algorithm. Initial velocities for the production runs were taken according to the Maxwell-Boltzmann distribution at 300 K. The pressure was held constant by

using the Parrinello-Rahman barostat.[69] The integration time step was set to 2 fs. The neighbor lists for non-bonded interactions were updated every 20 steps. Real-space electrostatic interactions were truncated at 1.2 nm. The van der Waals interactions were switched off between 1.0 and 1.2 nm, and short-range electrostatic interactions were cut off at 1.2 nm. All simulations were carried out using periodic boundary conditions for the simulation box and utilized the CHARMM36m[70,71] protein force field together with the CHARMM-modified[72] TIP3P water model and the CHARMM36 lipid parameters.[73] The openbabel web server[74] was used to create the input files for topology and parameter preparation of anle138b molecules with the CHARMM General Force Field (CGenFF 4.6).[75–78] The topologies were converted to GROMACS format using the cgenff charmm2gmx.py script (http://mackerell.umaryland.edu/charmm_ff.shtml#gromacs). All production simulations were preceded by a multi-step energy minimization and thermalization of the simulation systems. Atomic coordinates for analysis were stored every 100 ps.

## MD simulation: setup of full-length L1 Aβ40 model

To investigate lipid and small molecule binding to different parts of the lipidic polymorph L1 Aβ40 fibrils located by ssNMR measurements, we performed MD simulations of an Aβ40 fibril model in the presence of DMPG lipids. The cryo-EM structure of L1 Aβ40 fibrils (PDB-ID: 8ovk) was used to build a fibril model with two protofilaments, each composed of 25 helically arranged peptide chains. If not stated otherwise, the titratable groups of the protein structure were protonated according to their standard protonation states at pH 7. The L1 Aβ40 fibril structure was placed in a rectangular solvent box with the distance between the solute and the box edge of at least 2.5 nm (box dimensions: 9.6 nm x 8.2 nm x 12.3 nm). Based on this principal simulation system, two setups were built by adding the small molecule anle138b (representing both possible proton positions on the pyrazole ring; 1-nitrogen and 2-nitrogen in a 1:1 ratio):

a)  50 anle138b molecules were placed randomly in the solvent around the L1 Aβ40 model structure. In the next step, 700 DMPG molecules were placed randomly in the solvent around the L1 Aβ40 model structure. The final Aβ40 monomer to DMPG lipid to anle138b ratios were 1:14:1. Sodium and chloride ions were added to yield an ionic strength of 150 mM and to neutralize the net system charge. The final setup of this L1 Aβ40 model contained 408,900 atoms, including 99,900 water molecules. Ten independent NPT production simulations, each 1.0 µs long, were carried out for this simulation system.

b)  50 anle138b molecules were placed randomly in the solvent around the L1 Aβ40 model structure. Sodium and chloride ions were added to yield an ionic strength of 150 mM and to neutralize the net system charge. The final setup of this L1 Aβ40 model contained 448,600 atoms, including 138,700 water molecules. Ten independent NPT production simulations, each 0.25 µs long, were carried out for this simulation system.

## MD simulation: setup of truncated fibril model systems to study putative internal anle138b binding site

To investigate the anle138b binding to the loop region of Aβ40 fibrils, and the dynamics of internally bound anle138b, we performed MD simulations on a shorter, partially restrained substructure. Instead of simulating the whole fibril model, only peptide chain parts near the suspected internal small molecule binding sites were included (PDB-ID: 8OVK, residues 15-39). The truncated simulation model of the L1 Aβ40 fibrils was composed of 30 helically arranged peptide chains per protofilament. The truncated simulation model of L1 Aβ40 fibrils was placed in a rectangular solvent box with a distance between the solute and the box edge of at least 1.5 nm (box dimensions: 15.6 nm x 15.6 nm x 16.5 nm). The titratable groups of the protein structure were protonated according to their standard protonation states at pH 7. Acetyl

and N-methyl groups, respectively. Counterions (Na+, Cl−) were added to yield an ionic strength of 150 mM and to neutralize the net system charge.

For the initial configuration, one anle138b molecule was placed outside and above the central cavity of the loop region of L1 Aβ40 fibril, such that the long axis of the small molecule aligned with the fibril axis. For this simulation system, five independent NPT production simulations were carried out per possible proton position on the pyrazole nitrogens of anle138b, each 5.0 μs long.

An otherwise identical simulation with L1 Aβ40 fibrils featuring deprotonated lysine 28 side chains was performed for five independent NPT production simulations per possible pyrazole nitrogen protonation state of anle138b, each 5.0 μs long.

A flat-bottomed position restraint was used to keep the diffusion of the small molecule in the simulation box restricted to a cylindrical volume centered on the binding site of interest. The radius of the flat-bottom potential was chosen to correspond with the geometry of the ring-like cavity and the full extension of anle138b (radius 1 nm). This approach reduces computational cost significantly and increases the sampling efficiency of binding events to the cavity of the L1 Aβ40 fibril polymorph by preventing the compound from binding in a different site on the fibril surface. No biasing forces were applied to steer the ligand into the central cavity.

During all production MD simulations, the initial atomic coordinates of the Cα atoms of protein residues Asp1-Lys16 and Val36-Val40 were restrained using a harmonic potential with a force constant of 1000 kJ mol$^{-1}$ nm$^{-2}$ to ensure that the fibril models were stable, preserving the initial fold of the N- and C-terminal regions of the L1 Aβ40 fibril as resolved by cryo-EM avoiding undesired shearing and twisting of the protofilaments.

### MD simulation: reference simulations of anle138b in lipids
Anle138b conformations without β-sheet aggregates were set up and simulated as follows: A simulation box consisting of a solvated DMPG membrane patch composed of 76 lipid and 5266 water molecules and one anle138b ligand comprising 24,451 total atoms were simulated 6 ×1000 ns.

### MD simulation: analysis and visualization
The MD simulation trajectories were preprocessed and analyzed by tools provided by the GROMACS 2023 simulation software package[61,62], as well as custom-made bash, awk, and python scripts. The second half of the trajectory data were used for subsequent analyses as a measure to discard the initial equilibration phase and structural relaxation of the systems. Renderings of atomic coordinates were carried out with the PyMOL molecular visualization software[79] [http://www.pymol.org/pymol], and the 3D structural model was visualized using UCSF ChimeraX version 1.9.[47] Graphs and figures were created using Gnuplot 5.4 and Inkscape 1.3.

### MD simulation: contact analysis
Aβ40 fibril residues are quantified to be in contact with anle138b using the g_contacts and gmx mindist programs[80]. In accordance with the type of NMR experiments and the isotope labeling scheme, the following contact criteria were chosen: a) If the 1,2-N atoms of the anle138b molecule were found within a cut-off of 0.5 nm from any carbon or nitrogen atom of a protein residue, a contact was considered formed. b) In addition, a close interaction was, considered formed if all polar heavy atoms of anle138b (N, O, Br) molecules were found within a cut-off of 0.4 nm from all polar backbone or side chain heavy atoms of a protein residue. Contacts were averaged over time and over the individual trajectories of each simulation set. If not explicitly stated, only contacts to the core β-strand layers (neglecting the two β-strands on both protofilament edges) were considered for analysis.

### MD simulation: Density of atomic positions and interaction frequency analysis
Time-averaged 3D density maps were calculated using GROmaρs[81] separately for (i) all DMPG lipid heavy atoms, (ii) the carbon atoms of the lipid acyl chain, (iii) the phosphorous and oxygen atoms of the phosphate group, (iv) and the glycerol oxygen atoms. These grids represent the probability density of a molecule's position relative to the centered fibril structure. In brief, a grid of 0.1 nm in resolution, spanning the simulation box, was considered. The density was computed after least-square rigid-body fitting of their positions in the trajectories to the initial position of the L1 Aβ40 fibril structure.

The average density grids were calculated over all conformations of the last 100 ns of all MD simulations replicates. Density maps were visualized with PyMOL (The PyMOL Molecular Graphics System, Schrödinger, LLC) as isomesh contoured at 2σ.

Additionally, we calculated the average interaction frequencies for every amino acid with DMPG and anle138b. For this, we measured the minimum distance between any non-hydrogen atom of every amino acid of five layers from the center of each protofilament to (i) all heavy atoms of the phospholipids, (ii) the phosphate group of the phospholipids, (iii) the glycerol group of the phospholipids, (iv) any carbon atom of the acyl chains of the phospholipids, (v) any nitrogen and bromine atom of anle138b. An interaction was present if the distance was smaller than 5 Å. These interactions are normalized by the total number of frames, so that a value of 1.0 means "interaction always present", whereas a value of 0.0 means "interaction not existent". Again, the analysis focuses on the last 100 ns of all MD simulation replicates.

### Reporting summary
Further information on research design is available in the Nature Portfolio Reporting Summary linked to this article.

## Data availability
NMR spectra, titration curves, and all thermodynamic data are available in the Biological Magnetic Resonance Data Bank under accession code BMRB 53129. Backbone assignment data previously reported are available at BMRB 52006[9], as described in the Methods section. Cryo-EM density maps have been deposited in the Electron Microscopy Data Bank under accession codes EMD-53882 (pre-treatment fibril) and EMD-53880 (post-treatment fibril). The corresponding atomic models are available in the Protein Data Bank under accession codes PDB 9RAX (pre-treatment) and PDB 9RAW (post-treatment). Previously published atomic structures are available at PDB 6W0O[10], PDB 8FF3[11], and PDB 8OVK[9]. MD simulation data, including ligand topology and parameters, input and output files, and coordinates, are available in the Edmond repository [https://doi.org/10.17617/3.NRYUVQ].

Processed data from the control, pre-treatment, and post-treatment fibril experiments (ThT fluorescence, CD spectroscopy, ITC titration, and MD simulation analyses) are provided in the accompanying Source Data file. Source data are provided as a Source Data file. Source data are provided with this paper.

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

## Acknowledgements

This work was supported by the Max Planck Society (to CG) and the German Research Foundation (DFG) in the framework of the German Excellence Strategy-EXC 2067/1-390729940 (to CG) and the Emmy Noether Program to LBA (project number: 397022504). BF and GFS are grateful for the computational support and infrastructure provided by the Zentrum für Information und Medientechnologie (ZIM) at the Heinrich-Heine-Universität Düsseldorf and for the computing time supplied by the Forschungszentrum Jülich on the supercomputer JURECA-DC at the Jülich Supercomputing Center (JSC). We would like to thank Prof. Markus Zweckstetter for providing access to the Tecan Spark plate reader and the MicroCal PEAQ-ITC automated system.

## Author contributions

GFS, LBA, B.L.dG., and CG designed the project. MH, BF, GFS, LBA, and CG managed the project. KG, KO, and SB performed the protein expression and purification. AL and SR synthesized isotope-labeled anle138b. MH prepared the fibril samples. MH, EN, KX, and MS performed the ssNMR experiments. DR conducted the negative-staining EM analyses. CD prepared the cryo-EM grids and collected the cryo-EM images. BF processed the cryo-EM images, reconstructed the fibril structures, and built the atomic models; DM carried out the MD simulations. MH, BF, and DM visualized the results. GFS, LBA, B.L.dG., and CG supervised the project. MH, BF, DM, AG, B.L.dG., GFS, LBA, and CG wrote the original manuscript. All authors revised and edited the manuscript.

## Funding

## Competing interests

A. G. and C. G. are co-founders of MODAG. A.G. is a full-time employee of MODAG. A. L. and S.R. are partly employed by MODAG and are beneficiaries of the phantom share program of MODAG GmbH. A.L., S.R., C.G., and A.G. are co-inventors of WO/2010/000372. Anle138b is licensed by Teva Pharmaceutical Industries Ltd and is in clinical development in collaboration with MODAG. B.F. is an AstraZeneca employee. The remaining authors declare no competing interests.
