## [Transparent Peer Review file · Nature Communications]

Anle138b binds predominantly to the central cavity in lipidic A β ₄₀ fibrils and modulates fibril formation.

Corresponding Author: Professor Christian Griesinger

Version 0:

Reviewer comments:

Reviewer #1

(Remarks to the Author)

The work explores the binding of a small molecule drug anle138b to Abeta40 fibrils. The work provides convincing evidence to support their main argument that Anle138b binds inside the central cavity in “pre-treatment” fibrils and binds to the fibrillar surface in fibrils that have been fully formed.

This is a novel and interesting result, the work employs state-of-the-art approaches in ssNMR including ssNMR with signal enhancements by DNP, hence it demonstrates the power of the NMR-based methodology to address this type of question. For these reasons the work would be interesting to a wide audience of researchers.

The manuscript is well written and for the most part is easy to follow. The manuscript is appropriate for publication after adding a bit of clarity on a few minor issues:

- Line 63. “in vitro system developed in prior work”. It would be good to add a one-sentence clarification on what this system is.
- In SFig 2c – while the loop region forming the cavity is shown, for greater clarity it would help to add a label pointing to the cavity itself
- Line 102, “liposomes” are mentioned, better specify the lipid type, i.e “DMPG liposomes”
- SFig 1 shows the aggregation kinetics for various SMPR ratios. In the legend of SFig. 1 it would make sense to show “Anle138b:Ab40” rather than the other way around to match the SMPR parameter
- The “Methods” sections needs a bit more detail on how the ssNMR samples were prepared. In many works, fibrils are collected by pelleting using high-speed centrifugation, however, the “Methods” section does not mention that. Specifically, Line 112 mentions the 1D (H)N CP spectra. How was the ssNMR sample prepared so that CP signal strength serves as a measure of fibril formation?
- In line 112-113, negative stain EM is mentioned as a tool supporting fibril formation. Probably, you refer to somewhat higher density fibrils shown in Fig. 1b. of “post-treatment” and “control” samples. From our experience, the fibrillar coverage is not very uniform across the EM-grid, therefore, making a conclusion from a single image is not a very good idea. Please comment on why do you think that negative-stain EM is reliable in your situation.
- In SFig. 2a it makes sense to show the actual data rather than an illustration. SFig 1a does that; it is unclear why SFig 2a should be different.
- Fig. 4a caption says 3D (H)CANH ssNMR spectrum is shown. But the shown spectrum is looks more like 2D NCA. You may be showing a projection or a slice within a 3D spectrum, please clarify that.

Reviewer #2

(Remarks to the Author)

Comments on “The clinical drug candidate anle138b binds predominantly to the central cavity in lipidic A β ₄₀ fibrils and modulates fibril formation.”

This manuscript by Han et al. focuses on understanding the interactions between the small molecular aggregation inhibitor ANLE138b and fibrils of A-beta(1-40). The paper is overall clearly written and the data visualizations are effective. The

overall conclusion is that the small-molecule inhibitor binds predominantly in the fibril core when added before aggregation (“pre-treatment”) and predominantly to the fibril surface when added after fibril formation (“post-treatment”). The ssNMR experiments with isotope-labeled amino acids were particularly exciting in terms of developing insights into the specific interactions between drug and target.

However, in parts the data interpretation needs clarification to adequately support their conclusions. The MD simulations do not add that much to the reader’s confidence in the findings presented. There are also questions that need clarifying around the experimental design (detailed below). Taken as a whole, the findings are intriguing but a little equivocal, and should be of interest to the A-beta community.

My specific critiques are as follows:

1. For the post-treatment condition, the switch from DMPG to DLPG is not explained. The authors suggest that the DLPG “vesicles” (actually an unknown and ill-defined species) are acting purely as carriers for the drug. ITC data are presented to support a lack of interaction between DLPG and A-beta fibrils – however, ITC would not report on entropically-driven (as opposed to enthalpically-driven) binding. A lack of ITC signal doesn’t mean that there is no interaction, especially if predominantly enthalpic interactions (e.g., hydrophobic) are at play. A more established carrier like β -cyclodextrin would have been a better choice.
2. There is a lack of clarity around mass balance. The authors ought to measure how much peptide and how much drug remain in solution at the end of the control, pre-treatment, and post-treatment reactions. How much A-beta assembles into insoluble species, and how much drug co-aggregates with the insoluble states? Without this, it is not possible to understand the stoichiometry of drug:monomer within the fibrils.
3. On a related point, how do the authors explain the observed EM density within the fibril core in the control (0.3) and post-treatment (0.5) conditions? It is implausible that all of the pre-treatment density (0.7) should be assigned to drug if there is a similar contribution in these other conditions.
4. The design of the MD simulations needs more justification. Given that the authors are aiming to simulate lipidic fibrils, they should demonstrate that the conformation of DMPG around the fibrils approximates the ordering experimentally observed by EM. This is especially confusing given how similar the surface interactions of the drug are in the presence vs. absence of DMPG. Furthermore, the simulations with the truncated fibril models seem to be quite biased, with both backbone restraints on flanking residues and positional restraints on the drug molecule, and the drug aligned to the fibril axis. Where else could the drug go than into the central cavity? Finally, it is unclear what exactly the authors define as protonation states for Anle138b? I assume that they mean whether the proton is on the 1- or 2-nitrogen of the pyrazole ring (as opposed to a deprotonated state) but this should be clarified.
5. Figure legends seem to be missing from the main manuscript file.
6. On line 383, the citation should point to ref. 68.

Reviewer #3

(Remarks to the Author)

This is an interesting and well-written manuscript from the Griesinger lab. The authors explore the interaction of Anle138b — with A β 40 fibrils under a lipidic environment. Anle138b is a small molecule originally developed in their group and currently undergoing clinical trials for Parkinson’s disease and multiple system atrophy. The study employs an impressive suite of biophysical techniques, including cryo-EM, NMR, ITC, molecular dynamics (MD) simulations, and others, to dissect the binding behavior of Anle138b on both pre-formed and forming A β 40 fibrils. The key finding is that Anle138b binds within a central hydrophobic cavity of lipid-induced A β 40 fibrils (L1 polymorph), interacting with glycine and isoleucine residues, whereas in pre-formed fibrils, it binds more peripherally to solvent-exposed sites without altering the fibril core. These insights are valuable for guiding future rational design of amyloid-targeting therapeutics. I have a few suggestions to improve the manuscript:

(1) Throughout the manuscript, the term “lipidic A β ” is used, which may be misleading. In some contexts, this term could imply covalent linkage between lipids and A β . However, in this work, the authors refer to fibrils formed in the presence of lipids (specifically DMPG), not covalently modified A β . The authors should clarify this to avoid potential confusion.

(2) DMPG is used for co-incubation during fibril formation (pre-treatment), while DLPG is used to deliver Anle138b in post-treatment conditions. The rationale behind this choice should be explicitly addressed in the manuscript. Given that DMPG and DLPG differ in fluidity and fatty acid chain length, these lipids may influence the compound’s interaction with A β differently. Furthermore, it is not clear whether the pre-formed fibrils used in the post-treatment experiments were also generated in the presence of lipids or in their absence. Clarifying this will help contextualize the differences observed.

(3) The manuscript lacks information on the molar ratios of lipid to A β and lipid to Anle138b. These parameters are crucial, as varying lipid concentrations can significantly affect A β aggregation pathways and compound accessibility.

(4)The authors report a significant reduction in β -sheet content after 48 hours of incubation with Anle138b, based on CD spectra (Fig. S1C). However, the presented spectra only show modest shifts in the characteristic β -sheet peaks, as the positive peak slightly increases and negative peak value is a bit higher. To strengthen the conclusion, the authors should consider performing secondary structure deconvolution of the CD data to quantify the percentage of β -sheet content.

(5)NMR results (Fig. 1C) show a marked reduction in fibrillar signal upon Anle138b treatment in the pre-treatment condition, suggesting reduced fibril formation. In contrast, ssNMR in Fig.3 indicates that the compound can bind within the fibril core. These findings seem partially contradictory. Does Anle138b destabilize early fibril formation but remain stably bound in the formed fibril? Could the central binding pocket identified in figure 3 also play a role in modulating primary nucleation? Further discussion on how the structural data correlate with the dissolution or prevention of fibril formation would be valuable.

(6)To reinforce the functional relevance of the identified binding site, the authors might consider performing mutagenesis on key residues within the cavity (e.g., glycine or isoleucine residues) and assessing the effect on compound binding and fibril dissolution. This would provide mechanistic support for the role of the binding pocket in compound action against fibril formation.

(7)The conclusion that Anle138b competes with ThT for binding may be premature. ThT fluorescence can be influenced by multiple factors, including fibril structure and accessibility. To confirm direct competition, the authors could fix the compound concentration and titrate increasing amounts of ThT to observe potential displacement effects. Such data would help validate this claim.

(8)In the ITC experiments, the compound is solubilized in DLPG. It is unclear, however, whether the A β fibrils used in these experiments (post-treatment) were also prepared in DLPG or in a different lipid or buffer system. Since pre- and post-treatment experiments involve different lipids (DMPG vs. DLPG), this could affect the comparability of binding data and should be clarified.

(9)The authors apply a flat-bottomed positional restraint in their MD simulations. More explanation is needed regarding the purpose of this restraint and whether it might limit the natural flexibility of the compound-fibril interaction. Could this influence the binding mode or energetics observed?

Reviewer #4

(Remarks to the Author)

Version 1:

Reviewer comments:

Reviewer #1

(Remarks to the Author)

The authors have addressed all the minor issues that I've raised in the first review round. The manuscript should be good for publication.

Reviewer #2

(Remarks to the Author)

I appreciate the authors' responsiveness and clarity. However, their revisions leave a couple of issues that I believe still need to be addressed:

1. I was confused by the rationale that that DLPG was selected to reduce charge-based nonspecific interactions, given that DLPG and DMPG have the same head group and charge state. The subsequent parallel rationale based on membrane phase and fluidity, while speculative, is clearer and more plausible.

2. The initial version of the manuscript clearly explained that the DLPG is likely to be in a micelle-like state and is referred to as 'vesicles' only for convenience. This is an essential disclosure that has been lost in the revision, and I believe it needs to be added back to the manuscript.

Other than those caveats, in my opinion this is a really nicely executed and presented piece of work.

Reviewer #3

(Remarks to the Author)

The authors have satisfactorily addressed my concerns, and I recommend acceptance of the manuscript in its current form.

Reviewer #4

(Remarks to the Author)

REVIEWER COMMENTS

Reviewer #1 (Remarks to the Author):

The work explores the binding of a small molecule drug anle138b to Abeta40 fibrils. The work provides convincing evidence to support their main argument that Anle138b binds inside the central cavity in “pre-treatment” fibrils and binds to the fibrillar surface in fibrils that have been fully formed. This is a novel and interesting result, the work employs state-of-the-art approaches in ssNMR including ssNMR with signal enhancements by DNP, hence it demonstrates the power of the NMR-based methodology to address this type of question. For these reasons the work would be interesting to a wide audience of researchers. The manuscript is well written and for the most part is easy to follow. The manuscript is appropriate for publication after adding a bit of clarity on a few minor issues:

- Line 63. “in vitro system developed in prior work”. It would be good to add a one-sentence clarification on what this system is.

Good point. We have revised the manuscript to provide a clearer description of the in vitro system in response to your comment. The revisions are reflected in lines 63- 64 (highlighted in red) of the manuscript, as shown below.

Lines 62-66: Among these polymorphs, the type 1 lipidic A β ₄₀ fibril¹⁸⁻²⁰ is predominantly observed when the protein is allowed to aggregate in the presence of negatively charged DMPG lipid vesicles (lipid-to-protein molar ratio (LPR) = 30:1) at pH 6.5 and 37°C, as reported previously¹⁸. This fibril type, henceforth referred to as the L1 A β ₄₀ fibril, has been shown to resemble the pathological structures observed in the brains of patients.

- In SFig 2c – while the loop region forming the cavity is shown, for greater clarity it would help to add a label pointing to the cavity itself

We followed the suggestion and have added a label and arrow in the revised figure as recommended.

- Line 102, “liposomes” are mentioned, better specify the lipid type, i.e “DMPG liposomes”

We have changed it to “DMPG liposomes” and changed “liposomes” in line 108 to “DMPG liposome” for consistency highlighted.

- SFig 1 shows the aggregation kinetics for various SMPR ratios. In the legend of SFig. 1 it would make sense to show “Anle138b: Ab40” rather than the other way around to match the SMPR parameter

In line with your comment, we have revised the molar ratio notation in the SFig. 1 legend to match the SMPR definition (anle138b: A β ₄₀ molar ratio). We have also applied the same notation consistently in the legends of Main Figures 1, 3, 4, 5, and Supplementary Figures 1, 3, 4, 5, 11, 12, 13, and 16. All revisions are reflected in the new manuscript with changes highlighted in red.

- The “Methods” sections needs a bit more detail on how the ssNMR samples were prepared. In many works, fibrils are collected by pelleting using high-speed centrifugation, however, the “Methods” section does not mention that. Specifically, Line 112 mentions the 1D (H)N CP spectra. How was the ssNMR sample prepared so that CP signal strength serves as a measure of fibril formation?

We have revised the 'Methods' section to clarify this point. Specifically, we have added the information that both pre-treatment and post-treatment fibrils were collected by high-speed centrifugation prior to ssNMR measurements. This information has been added in red at lines 469-472 (**highlighted in red**) of the revised manuscript, as shown below.

Lines 466- 472: The anle138b-loaded DLPG vesicles were used to treat preformed L1 A β ₄₀ fibrils for subsequent ITC, DNP-NHHC, HCANH titration, **ssNMR**, and Cryo-EM experiments. With the exception of the ITC experiment, L1 A β ₄₀ fibrils were incubated at 37 °C for 1 hour to allow sufficient interaction prior to measurement. **The fibrils were collected by high-speed centrifugation at 55,000 g for 60 minutes for subsequent ssNMR measurements. The pelleted fibrils were then packed into MAS rotors using a funnel consisting of a trimmed pipette tip and a tabletop centrifuge.**

- In line 112-113, negative stain EM is mentioned as a tool supporting fibril formation. Probably, you refer to somewhat higher density fibrils shown in Fig. 1b. of “post-treatment” and “control” samples. From our experience, the fibrillar coverage is not very uniform across the EM-grid, therefore, making a conclusion from a single image is not a very good idea. Please comment on why do you think that negative-stain EM is reliable in your situation.

We agree that negative-stain EM images should not be used as an independent quantitative tool. In this manuscript, they formed part of a cross-validation process alongside NMR, ThT fluorescence analysis, and CD

spectra, all of which were obtained from the same sample. This is specified in the legends of Supplementary Figures 3 (panels a and b) and 5, thereby strengthening the reliability of our conclusions regarding fibril formation and quantity.

- In SFig. 2a it makes sense to show the actual data rather than an illustration. SFig 1a does that; it is unclear why SFig 2a should be different.

We chose to present Supplementary Figure 2a as a schematic to provide a clear summary of the aggregation kinetics before and after anle138b administration. The corresponding experimental data are shown in Supplementary Figure S1a (for pre-aggregation) and Figure 1c (for both pre- and post-aggregation). Unlike Supplementary Figure 1a, which presents raw kinetic traces, Supplementary Figure 2a is intended to give readers a conceptual overview of the experimental timeline and treatment point. We have revised the figure legend accordingly to clarify this purpose. This has been **highlighted in red** for your reference.

Supplementary Figure 2 | Schematic illustration of experiments investigating the binding sites of anle138b on L1 A β ₄₀ fibrils.

a. Schematic illustration of L1 A β ₄₀ fibril formation under pre-treatment and post-treatment conditions. In the post-treatment condition, anle138b (blue) is added after fibril formation. In the pre-treatment condition, fibrils form in the presence of anle138b (red), which extends the lag phase and reduces the final fibril amount. The two arrows indicate the time points at which anle138b was administered during the fibril formation process. Blue represents the post-treatment condition, and red represents the pre-treatment condition. **This schematic provides a conceptual overview of the experimental timeline and administration of anle138b, complementing the corresponding kinetic data shown in Supplementary Fig. 1a and Fig. 1c.**

- Fig. 4a caption says 3D (H)CANH ssNMR spectrum is shown. But the shown spectrum is looks more like 2D NCA. You may be showing a projection or a slice within a 3D spectrum. Please clarify that.

Thank you for your comment. The spectrum shown in Fig. 4a is indeed a 2D projection of the 3D (H)CANH ssNMR spectrum. We have clarified this in the figure legend to avoid confusion, and the corresponding revision has been **highlighted in red** in the updated manuscript.

Fig. 4: Titration of anle138b binding to L1 A β ₄₀ fibrils under post-treatment conditions.

a. 3D (H)CANH solid-state NMR spectra of L1 A β ₄₀ fibrils titrated with increasing concentrations of anle138b (**anle138b: A β ₄₀ molar ratio (SMPR) from 0 to 1.2**). **The spectrum shown is a 2D projection of the 3D spectrum.** Chemical shift perturbations (CSPs) and signal intensity reductions are observed upon treatment with anle138b. Residues exhibiting both CSPs and signal attenuation are marked in pink arrows; those showing CSPs only are marked in gray arrows. Peaks for residues K16, D23, S26, and K28 (highlighted in purple) become undetectable at the highest anle138b concentration.

Reviewer #2 (Remarks to the Author):

Comments on “The clinical drug candidate anle138b binds predominantly to the central cavity in lipidic A β ₄₀ fibrils and modulates fibril formation.”

This manuscript by Han et al. focuses on understanding the interactions between the small molecular aggregation inhibitor ANLE138b and fibrils of A-beta(1-40). The paper is overall clearly written and the data visualizations are effective. The overall conclusion is that the small-molecule inhibitor binds predominantly in the fibril core when added before aggregation (“pre-treatment”) and predominantly to the fibril surface when added after fibril formation (“post-treatment”). The ssNMR experiments with isotope-labeled amino acids were particularly exciting in terms of developing insights into the specific interactions between drug and target. However, in parts the data interpretation needs clarification to adequately support their conclusions. The MD simulations do not add that much to the reader’s confidence in the findings presented. There are also questions that need clarifying around the experimental design (detailed below). Taken as a whole, the findings are intriguing but a little equivocal, and should be of interest to the A-beta community.

Thank you for the overall positive assessment. We are well aware, as pointed out, that the MD simulations have particular limitations, for example, in the timescale that is feasible to simulate. Still, they add value to the manuscript, since the snapshots from the MD confirm the physical ‘reasonableness’ of the proposed binding interactions, and also help to clarify possible binding poses.

My specific critiques are as follows:

1. For the post-treatment condition, the switch from DMPG to DLPG is not explained. The authors suggest that the DLPG “vesicles” (actually an unknown and ill-defined species) are acting purely as carriers for the drug. ITC data are presented to support a lack of interaction between DLPG and A-beta fibrils – however, ITC would not report on entropically-driven (as opposed to enthalpically-driven) binding. A lack of ITC signal doesn’t mean that there is no interaction, especially if predominantly enthalpic

interactions (e.g., hydrophobic) are at play. A more established carrier like β -cyclodextrin would have been a better choice.

We appreciate the reviewer's valuable comment. In the post-treatment condition, DLPG was used to deliver anle138b directly to pre-formed L1 A β ₄₀ fibrils. These fibrils are structurally identical to those used in the pre-treatment and control conditions, as all were formed under the same DMPG-containing conditions. In these cases, DMPG was used as a lipid that promotes fibril formation, and both cryo-EM density maps and ssNMR NOE analyses confirmed that DMPG is bound to the fibril surface¹⁸. The structural characteristics of DMPG binding are described in greater detail in response to Reviewer #3's Comment 1 and have been addressed in the revised manuscript (Lines 75–77, highlighted in blue).

DMPG carries a negative charge and engages in hydrophobic interactions with the fibril surface. To avoid such charge-based interactions involving the DMPG head group, DLPG was selected as an alternative. DLPG also carries a negative charge but has a lower melting temperature ($T_m = -3\text{ }^\circ\text{C}$), which results in high membrane fluidity at the experimental temperature (25 $^\circ\text{C}$). This property facilitates the dispersion and release of hydrophobic molecules such as anle138b. Therefore, DLPG provides a favorable environment for solubilizing and delivering the drug while minimizing nonspecific binding to A β ₄₀ fibrils. This has been reflected in the revised manuscript (Lines 453–459, highlighted in green).

Lines 453–459: For post-treatment conditions, DLPG vesicles were used to deliver anle138b to pre-formed L1 A β ₄₀ fibrils. DLPG was selected instead of DMPG to reduce potential charge-based nonspecific interactions, as DLPG headgroups are less prone to stable association with the fibril surface. Additionally, DLPG has a lower melting temperature ($T_m = -3\text{ }^\circ\text{C}$), which ensures high membrane fluidity at 25 $^\circ\text{C}$ and facilitates dispersion of hydrophobic molecules such as anle138b. To minimize co-sedimentation artifacts during centrifugation, small unilamellar DLPG vesicles were used.

No detectable interaction between DLPG and L1 A β ₄₀ fibrils was observed using three independent methods: ITC, ssNMR NOE, and cryo-EM. Acknowledging that ITC alone may not capture all types of interactions, we employed additional analyses using ssNMR NOE and cryo-EM to assess the potential binding between DLPG and A β ₄₀ fibrils comprehensively. All three techniques consistently failed to detect any interaction signals, thereby compensating for the limitations of any single method. *Together, these findings confirm that DLPG served as a non-interacting delivery vehicle in our system.* This has been included in the revised manuscript (Lines 230–233, highlighted in green).

Lines 230–233: DLPG vesicles lacking anle138b produced no measurable heat signal in ITC (Supplementary Fig. 13b, Supplementary Table 2). This absence of detectable interaction was corroborated by ssNMR NOE and cryo-EM, which showed no DLPG-specific contacts or densities on the L1 A β ₄₀ fibril surface (Fig. 2b, Fig. 5).

We appreciate the reviewer's suggestion to use β -cyclodextrin as a potential delivery vehicle. Unfortunately, we have not been able to solubilize Anle138b in α -, β -, or γ -cyclodextrin. Furthermore, previous studies have shown that β -cyclodextrin can itself influence A β aggregation by inhibiting fibril formation or

disassembling pre-formed fibrils¹ (Shinde et al., 2017). Moreover, β -cyclodextrin has been reported to form inclusion complexes with amyloid peptides themselves^{2,3} (Yang et al., 2012; Qin et al., 2002), which could further complicate the interpretation of anle138b's independent effects on fibril formation. Since the objective of this study was to isolate the effects of anle138b alone, β -cyclodextrin was intentionally excluded.

1. Shinde, Meenakshi N., et al. "Sulfobutylether- β -cyclodextrin for inhibition and rupture of amyloid fibrils." *The Journal of Physical Chemistry C* 121.36 (2017): 20057-20065.
2. Wahlstrom, Anna, et al. "Specific binding of a β -cyclodextrin dimer to the amyloid β peptide modulates the peptide aggregation process." *Biochemistry* 51.21 (2012): 4280-4289.
3. Qin, Xu-rong, Hiroshi Abe, and Hiroshi Nakanishi. "NMR and CD studies on the interaction of Alzheimer β -amyloid peptide (12–28) with β -cyclodextrin." *Biochemical and biophysical research communications* 297.4 (2002): 1011-1015.

2. There is a lack of clarity around mass balance. The authors ought to measure how much peptide and how much drug remain in solution at the end of the control, pre-treatment, and post-treatment reactions. How much A-beta assembles into insoluble species, and how much drug co-aggregates with the insoluble states? Without this, it is not possible to understand the stoichiometry of drug:monomer within the fibrils.

We appreciate the reviewer's valuable comment.

Using CD spectroscopy and solution-state NMR, we confirmed that under both control and post-treatment (pre-formed fibril) conditions, $A\beta_{40}$ monomers were completely converted into insoluble fibrils. In the pre-treatment condition, the CD spectra showed the disappearance of random coil signals and the emergence of β -strand signatures. Additionally, ssNMR INEPT HN-H spectra did not show signals corresponding to free monomers or unstructured peptide states. The data collectively support that $A\beta_{40}$ is predominantly found in the fibrillar form under both control and post-treatment conditions. Supplementary Figure 3 panels c and d include additional ThT fluorescence and CD spectrum data from the supernatant after fibril formation.

Regarding drug binding, ITC analysis showed no significant heat change between DLPG vesicles and $A\beta_{40}$ fibrils, whereas a clear exothermic signal was observed between anle138b-loaded DLPG vesicles and $A\beta_{40}$ fibrils. This supports that anle138b directly interacts with the fibrils. Furthermore, ssNMR titration experiments performed under the same conditions revealed a similar stoichiometry, further supporting the interaction between the drug and fibrils.

Moreover, 1D solution-state ¹H NMR analysis of the supernatant from post-treatment samples (after ultracentrifugation) revealed no detectable anle138b signals at SMPR ratios of 0.2:1 and 0.8:1, whereas ~30% of free drug remained at a ratio of 1.2:1. These results suggest that, under these conditions, the majority of anle138b is incorporated into fibrils. The corresponding 1D ¹H NMR spectra are shown in Supplementary Figures 21 and

22 and have been reflected in the revised manuscript (Lines 127–133, Lines 220–233, Lines 375–377; highlighted in green).

Lines 127–134 This observation was supported by analysis of the supernatant following ultracentrifugation: CD and ThT fluorescence spectra showed that β -strand-like species remained in the supernatant under pre-treatment conditions, indicating the presence of soluble, non-fibrillar $A\beta_{40}$ aggregates (Supplementary Fig. 3c, d). In contrast, these species were absent in the control or post-treatment samples, where nearly all $A\beta_{40}$ sedimented as fibrils (Supplementary Fig. 3c, d), i.e. post-treatment resulted in no significant change in fibril quantity (Fig. 1c, Supplementary Fig. 3b).

Lines 220–233: CD spectroscopy and ThT fluorescence confirmed full conversion of $A\beta_{40}$ monomers into fibrils under both control and post-treatment conditions, consistent with near-complete sedimentation of L1 $A\beta_{40}$ after ultracentrifugation (Supplementary Fig. 3).

To further assess drug incorporation, we analyzed the supernatant after ultracentrifugation using 1D solution NMR. No free anle138b signal was detected at anle138b: $A\beta_{40}$ ratios of 0.2:1 and 0.8:1, while ~30% of the drug remained unbound at a 1.2:1 ratio (Supplementary Figs. 21, 22). These results align with the stoichiometry measured by ITC (Fig. 4b, Supplementary Table 2) and NMR titration (Fig. 4a). Together, they indicate that the majority of anle138b associates with the fibrils.

DLPG vesicles lacking anle138b produced no measurable heat signal in ITC (Supplementary Fig. 13b, Supplementary Table 2). This absence of detectable interaction was corroborated by ssNMR NOE and cryo-EM, which showed no DLPG-specific contacts or densities on the L1 $A\beta_{40}$ fibril surface (Fig. 2b, Fig. 5).

Lines 375–377: These results are further supported by ultracentrifugation analysis of the supernatant under post-treatment conditions, confirming near complete conversion of $A\beta_{40}$ into fibrils and a substantial incorporation of anle138b into the fibril fraction.

Given the lipidic nature of the fibrils used in this study, it is technically challenging to distinguish whether the drug is bound to the fibril core or to the surrounding lipid assemblies using conventional mass-based quantification methods. Therefore, the quantification of drug–fibril interactions was based on ITC and ssNMR, which represent the most reliable and informative approaches available for this system. We acknowledge that further development of complementary techniques will be necessary to determine the absolute amount and site-specificity of drug binding with greater resolution in future studies.

Accordingly, we focused on ITC and NMR titration experiments to provide the most quantitative and reliable evaluation of drug–fibril interactions and their stoichiometry within the scope of this study.

3. On a related point, how do the authors explain the observed EM density within the fibril core in the control (0.3) and post-treatment (0.5) conditions? It is implausible that all of the pre-treatment density (0.7) should be assigned to drug if there is a similar contribution in these other conditions.

We indeed don't know what the density is in the control fibrils and therefore refrained from a detailed discussion on potential water or side chains occupying this space. Clearly, in the control but also in the post-treatment condition, the contribution from anle138b is not sufficient to induce peaks in the DNP-NHHC experiments. Those peaks, which directly confirm that anle138b binds within the fibril core, were only observed under the pre-treatment condition. Based on these results, we interpret the increased core density in the pre-treatment sample as reflecting more anle138b binding in the cavity than in the post-treatment sample. While we do not attribute the entire 0.7 density solely to the drug, the relative increase, compared to control and post-treatment, supports the presence of anle138b in the core.

4. The design of the MD simulations needs more justification. Given that the authors are aiming to simulate lipidic fibrils, they should demonstrate that the conformation of DMPG around the fibrils approximates the ordering experimentally observed by EM. This is especially confusing given how similar the surface interactions of the drug are in the presence vs. absence of DMPG. Furthermore, the simulations with the truncated fibril models seem to be quite biased, with both backbone restraints on flanking residues and positional restraints on the drug molecule, and the drug aligned to the fibril axis. Where else could the drug go than into the central cavity? Finally, it is unclear what exactly the authors define as protonation states for Anle138b? I assume that they mean whether the proton is on the 1- or 2-nitrogen of the pyrazole ring (as opposed to a deprotonated state) but this should be clarified.

We thank the reviewer for this suggestion. The MD simulation methods section has been revised accordingly to include the relevant details (highlighted in green).

- DMPG Conformation and Comparison to cryo-EM and ssNMR Data

We've enhanced our analysis of the MD simulations by incorporating average density grids and interaction frequencies to more thoroughly characterize the DMPG conformation around the L1 fibrils (Supplementary Figure 23, see MD simulation methods section for details). We acknowledge that the simulations, performed at 300 K, may not fully replicate the lipid ordering seen in cryo-EM. However, the lipid-binding regions

and lipid-fibril interactions, as found in the cryo-EM and ssNMR experiments depicted in Figures 2 and 5, are well-approximated. Furthermore, the interatomic contacts of surface-bound anle138b (Supplementary Fig. 18) reveal significant differences in the distributions of minimum distances between anle138b pyrazole nitrogen atoms and all N or C atoms of residues I31 and I32, both with and without DMPG. Notably, we observed no close I32–anle138 b contacts in the presence of lipids, suggesting competition of lipid and anle138b binding at the L1 A β ₄₀ fibril surface near the position of Ile32. This has been included in the revised manuscript (Lines 294–295, highlighted in green).

Lines 291–296: Notably, anle138b did not bind to the central cavity in either lipid-containing or lipid-free simulations. Lipid-free simulations show short distances (below 5 Å) between the pyrazole NH of anle138b and C δ of residue Ile32, consistent with NMR data, while the lipid-containing simulations do not, suggesting competition of lipid and anle138b binding at the L1 A β ₄₀ fibril surface near the position of Ile32 (Fig. 6b, c, Supplementary Fig.18a, b).

● Use of Restraints and Sampling of Central Cavity Binding

The central, ringlike cavity located along the fibril axis is narrow with a maximal diameter of approximately 1 nm. We carried out MD simulations on a truncated fibril model specifically designed to answer the questions a) is anle138b (sterically) able to bind to the central cavity, b) dynamic inside the fibril binding site, and c) reproducing the interatomic protein-ligand contacts observed in the NMR experiments. To remove the possible bias from initial configurations, we opted to start from ligand positions outside of the fibril with no fibril-ligand contacts, instead of, e.g., a docked pose inside the central cavity. No biasing forces were applied to steer the ligand into the central cavity. To keep the diffusion of the small molecule in the simulation box restricted to a cylindrical volume centered on the binding site of interest, the radius of the flat-bottom potential was chosen to correspond with the geometry of the ring-like cavity and the full extension of anle138b. This approach reduces computational cost significantly and increases the sampling efficiency of binding events to the cavity of the L1 A β ₄₀ fibril polymorph by preventing the compound from binding in a different site on the fibril surface. The flat-bottom potential, however, does not impede the compound to freely rotate, orient, un- and rebind from the fibril (cavity); it results in no forces when the molecule is bound, or just outside the entrance of the cavity. The orientational alignment of unbound anle138b is shown in Supplementary Figure 24.

During the production MD simulations of the truncated fibrils, the initial atomic coordinates of the C α atoms of protein residues Gln15-Lys16 and Val36-Val40 were restrained using a harmonic potential with a force constant of 1000 kJ mol⁻¹ nm⁻² to ensure that the fibril models were stable, preserving the initial fold of the N- and C-terminal regions of the L1 A β ₄₀ fibril as resolved by cryo-EM. The restrained atoms mainly serve to avoid undesired shearing and twisting of the protofilaments and are more than 1.5 nm away from the residues located near and around the central cavity, while the rest of all protein atoms are free from restraining forces. While the chosen design does restrict the conformational space of the ligand such that the simulations will only capture a subset of the possible binding sites, i.e. the central cavity and the tip of the fibril, the simulations do not bias towards or reinforce a specific pose. See also reply to comments (9) of **Reviewer #3 on p. 16**. We revised the Method Section to improve the clarity regarding the setup and implications of the use of flat-bottomed potentials in the simulations with truncated fibril models.

- Protonation states for Anle138b

Indeed, both proton positions (1-nitrogen and 2-nitrogen) were simulated. The nomenclature in the Method Section has been adjusted such that the description is not confused with protonated vs deprotonated states of the pyrazole ring. (Lines 624-625, Lines 652–654; highlighted in green).

Lines 623–625: Based on this principal simulation system, two setups were built by adding the small molecule anle138b (representing both possible proton positions on the pyrazole ring; 1-nitrogen and 2-nitrogen in a 1:1 ratio):

Lines 650–654: For the initial configuration, one anle138b molecule was placed outside and above the central cavity of the loop region of L1 A β ₄₀ fibril, such that the long axis of the small molecule aligned with the fibril axis. For this simulation system, five independent NPT production simulations were carried out per possible proton position on the pyrazole nitrogens of anle138b, each 5.0 μ s long.

5. Figure legends seem to be missing from the main manuscript file.

We thank the reviewer for the comment. In the revised manuscript, all figure legends have been clearly included in the main file. We will make sure to check such details more carefully in future submissions.

6. On line 383, the citation should point to ref. 68.

Thank you for your comment. We have corrected the citation to refer to Ref. 68 (Green color), and this change has been implemented on line 386 of the revised manuscript.

Reviewer #3 (Remarks to the Author):

This is an interesting and well-written manuscript from the Griesinger lab. The authors explore the interaction of Anle138b —with A β 40 fibrils under a lipidic environment. Anle138b is a small molecule originally developed in their group and currently undergoing clinical trials for Parkinson's disease and multiple system atrophy. The study employs an impressive suite of biophysical techniques, including cryo-EM, NMR, ITC, molecular dynamics (MD) simulations, and others, to dissect the binding behavior of Anle138b on both pre-formed and forming A β 40 fibrils. The key finding is that Anle138b binds within a central hydrophobic cavity of lipid-induced A β 40 fibrils (L1 polymorph), interacting with glycine and isoleucine residues, whereas in pre-formed fibrils, it binds more peripherally to solvent-exposed sites without altering the fibril core. These insights are valuable for guiding future rational design of amyloid-targeting therapeutics. I have a few suggestions to improve the manuscript:

(1) Throughout the manuscript, the term “lipidic A β ” is used, which may be misleading. In some contexts, this term could imply covalent linkage between lipids and A β . However, in this work, the authors refer to fibrils formed in the presence of lipids (specifically DMPG), not covalently modified A β . The authors should clarify this to avoid potential confusion.

We thank the reviewer for the valuable comments. To avoid any misunderstanding that the term “lipidic A β ” implies covalent linkage between lipids and A β , we have revised the manuscript to clarify that the fibrils were formed in the presence of DMPG, with **non-covalent** lipid interactions. The revisions are reflected in *lines 75–77 (highlighted in blue)* of the manuscript, as shown below.

Lines 74-77: This cavity is centrally located along the fibril axis and is hereafter referred to as the central cavity (Supplementary Fig. 2b, c). The L1 A β ₄₀ fibril is formed in the presence of DMPG liposomes, where lipid acyl chains interact with hydrophobic surfaces of the fibril (L17–F19 and A30–V36) through non-covalent interactions¹⁸.

(2) DMPG is used for co-incubation during fibril formation (pre-treatment), while DLPG is used to deliver Anle138b in post-treatment conditions. The rationale behind this choice should be explicitly addressed in the manuscript. Given that DMPG and DLPG differ in fluidity and fatty acid chain length, these lipids may influence the compound's interaction with A β differently. Furthermore, it is not clear whether the pre-formed fibrils used in the post-treatment experiments

were also generated in the presence of lipids or in their absence. Clarifying this will help contextualize the differences observed.

We thank the reviewer for the comment. This comment regarding DLPG and DMPG overlaps with the first comment from Reviewer #2, and the related explanation and experimental data were provided in that response.

Hydrophobic compounds such as anle138b generally have low aqueous solubility. Forming complexes with cyclodextrin or serum albumin is one of the strategies used to increase the solubility of such compounds. In this study, DLPG was used for solubilization and delivery of anle138b.

The system is based on fibrils formed in the presence of DMPG, which interacts with the L1 A β_{40} fibril surface through hydrophobic interactions. In the post-treatment condition, DLPG was used to deliver anle138b. DLPG has high membrane fluidity at room temperature and shows less nonspecific interaction via charge-based effects with L1 A β_{40} fibrils formed in the presence of DMPG.

Under the experimental conditions, DMPG tended to form relatively large vesicles compared to DLPG. These larger vesicles could affect sample recovery through ultracentrifugation in ssNMR experiments, and therefore, DLPG was used in the post-treatment condition. To match the conditions used in ssNMR titration, DLPG was also used in the ITC experiment.

The fibril formation conditions used in the post-treatment experiments are described in lines 125–126 of the manuscript (highlighted in blue).

Lines 121-126: To evaluate the effect of anle138b at different stages of fibril formation, we compared three experimental conditions: 1) pre-treatment (SMPR = 1.2, with anle138b added before initiating fibril formation), 2) post-treatment (SMPR = 1.2, with anle138b applied after fibril formation was completed), and 3) control (fibril without anle138b) (Supplementary Fig. 2a). Pre-formed fibrils used in the post-treatment condition were generated with DMPG liposomes under identical conditions to the control fibrils.

(3) The manuscript lacks information on the molar ratios of lipid to A β and lipid to Anle138b. These parameters are crucial, as varying lipid concentrations can significantly affect A β aggregation pathways and compound accessibility.

We thank the reviewer for this helpful comment. We have revised the manuscript to include the molar ratio of lipid to A β_{40} (LPR = 30:1) in the description of the in vitro system, as reflected in lines 63–64 (highlighted in red), and previously noted in our response to Reviewer #1 (1st comment). Regarding the molar ratio of lipid to Anle138b, we have added this information to the Methods section. Specifically, the A β_{40} : DMPG: anle138b molar ratios have been included in the “L1 A β_{40} fibril preparation” and “Aggregation assay” subsections to clarify experimental conditions. The main text and figure legends refer to SMPR values to maintain consistency in data interpretation, while the full molar ratios are now provided in the Methods section for reference. These revisions

are reflected in lines 63-64 (highlighted in red), 442-444, and 476 (highlighted in blue) of the manuscript, as shown below.

Lines 62-66: Among these polymorphs, the type 1 lipidic A β ₄₀ fibril¹⁸⁻²⁰ is predominantly observed when the protein is allowed to aggregate in the presence of negatively charged DMPG lipid vesicles (lipid-to-protein molar ratio (LPR) = 30:1) at pH 6.5 and 37°C, as reported previously¹⁸. This fibril type, henceforth referred to as the L1 A β ₄₀ fibril, has been shown to resemble the pathological structures observed in the brains of patients.

Lines 440-445: For fibril formation experiments, the DMPG film was hydrated and sonicated in 10 mM sodium phosphate buffer (pH 6.5) for 5 minutes to generate vesicles that promote A β ₄₀ aggregation. Under the pre-treatment fibril condition, anle138b was incorporated into DMPG during lipid film preparation to produce vesicles containing both lipid and anle138b (LPR = 30:1, SMPR = 0.2-1.2). Control fibrils and pre-treatment fibrils were formed with DMPG liposomes with or without anle138b.

Lines 474-476: Aggregation assays were conducted under the same lipidic conditions used for fibril formation, either with DMPG liposomes (SUV) alone (control fibril condition) or with DMPG liposomes (SUV) containing anle138b (LPR = 30:1, SMPR = 0.6 or 1.2).

(4) The authors report a significant reduction in β -sheet content after 48 hours of incubation with Anle138b, based on CD spectra (Fig. S1C). However, the presented spectra only show modest shifts in the characteristic β -sheet peaks, as the positive peak slightly increases and negative peak value is a bit higher. To strengthen the conclusion, the authors should consider performing secondary structure deconvolution of the CD data to quantify the percentage of β -sheet content.

We appreciate the reviewer's thoughtful feedback.

Supplementary Fig. 1C presents kinetic data monitored in real-time using ThT fluorescence with a Tecan plate reader. After 48 hours of monitoring ThT-containing samples with the plate reader, the same samples were immediately analyzed by CD spectroscopy, and the corresponding results are shown in Supplementary Fig. 1C.

To further clarify β -sheet formation and secondary structural characteristics, we performed additional analyses on the same samples prepared at the highest concentration of Anle138b (SMPR=1.2). These analyses involved various methods including CD spectroscopy, ThT fluorescence, ssNMR, and cryo-EM, and the comprehensive structural evaluations are presented in Fig. 1, 2, and Supplementary Fig. 3 and 5.

The CD spectra shown in Supplementary Fig. 3 exhibit typical β -sheet curves under all conditions; however, samples treated with anle138b during fibril formation displayed reduced signal intensity. We interpret this as reflecting a reduction in fibril quantity rather than structural alterations in the fibrils themselves.

L1 A β ₄₀ fibril structure contains an extended loop region (Ala21-Gly33) between β 2 and β 3, comprising non-canonical structural elements such as β -turns and loops. Recent literature¹ (DichroIDP, 2023) suggests that accurate quantitative secondary structure analysis by conventional CD deconvolution methods is challenging for

proteins containing such non-canonical structures. Moreover, Khrapunov (2009) reported that CD analysis reliably quantifies secondary structures only for repetitive all- α or canonical all- β proteins, whereas non-canonical elements such as β -turns, loops, or distorted β -strands can lead to misleading interpretations².

Given these structural characteristics of our fibrils and the presence of non-canonical β -turns and loops, we decided against performing secondary structure deconvolution of the CD spectra in this study. We have also revised the main text (lines 112–119, highlighted in blue) to clarify this point.

Lines 109-120: CD spectra showed β -sheet signatures under all conditions at the beginning of incubation after mixing A β ₄₀ with DMPG liposomes (Supplementary Fig. 1b), indicating similar initial secondary structures regardless of the presence of anle138b. After 48 hours of incubation, a strong β -sheet signal was retained in the control sample (SMPR = 0), whereas β -sheet formation was reduced in samples treated with anle138b (SMPRs = 0.6 and 1.2). Fibril formation was most reduced at SMPR 1.2, notably, indicating a dose dependence (Supplementary Fig. 1a, c). These results were further supported by two complementary approaches performed on a dedicated fibril sample (pre-treatment, post-treatment, control fibril): 1D (¹H)¹⁵N CP spectrum signal intensity, and quantification of fibrils by negative stain EM and CD spectroscopy (Fig. 1b, c, Supplementary Fig. 3). All three methods consistently indicated a dose-dependent inhibition of fibril formation by anle138b.

1. Miles, A. J., Drew, E. D., & Wallace, B. A. (2023). DichroIDP: a method for analyses of intrinsically disordered proteins using circular dichroism spectroscopy. *Communications Biology*, 6(1), 823.
2. Khrapunov, S. (2009). Circular dichroism spectroscopy has intrinsic limitations for protein secondary structure analysis. *Analytical biochemistry*, 389(2), 174-176.

(5) NMR results (Fig. 1C) show a marked reduction in fibrillar signal upon Anle138b treatment in the pre-treatment condition, suggesting reduced fibril formation. In contrast, ssNMR in Fig.3 indicates that the compound can bind within the fibril core. These findings seem partially contradictory. Does Anle138b destabilize early fibril formation but remain stably bound in the formed fibril? Could the central binding pocket identified in figure 3 also play a role in modulating primary nucleation? Further discussion on how the structural data correlate with the dissolution or prevention of fibril formation would be valuable.

At the current stage, it is difficult to definitively conclude that anle138b directly affects primary nucleation via interaction with the central cavity, based solely on structural data obtained from mature fibrils. It remains possible that anle138b occupies the central cavity during the fibril formation process, or alternatively, inserts into pre-formed fibrils. To explore this possibility, we examined the post-treatment condition, in which anle138b was added to pre-formed L1 A β ₄₀ fibrils and incubated for 1 hour. Under these conditions, ssNMR analysis did not provide conclusive evidence of anle138b insertion into the central cavity. These results suggest that compound insertion may vary depending on the timing of exposure during the fibril formation process. This

apparent dual mode of action is further discussed in the final section of the Discussion (lines 403–414), where we propose that anle138b both inhibits early-stage aggregation and selectively binds to mature fibrils.

Lines 403-414: Our results reveal a dual mode of action: inhibition of fibril formation during early aggregation and selective binding to mature fibrils without major structural disruption. From a thermodynamic perspective, it is intriguing that anle138b can stably associate with fibrils while simultaneously impeding their formation. Although the structural basis for the observed inhibition remains unresolved, one plausible explanation is that anle138b preferentially stabilizes early, non-fibrillar aggregates, thereby reducing the formation of mature fibrils. This mechanism is similar to that proposed for anle145c, a structurally related diphenyl-pyrazole (DPP) compound, which has been shown to inhibit hIAPP fibrillation and has been proposed to stabilize non-toxic oligomeric species through a thermodynamically driven process⁶⁸. Moving forward, we aim to identify and structurally characterize these compound-stabilized early intermediates, which may hold the key to understanding the therapeutic mechanism of anle138b.

(6) To reinforce the functional relevance of the identified binding site, the authors might consider performing mutagenesis on key residues within the cavity (e.g., glycine or isoleucine residues) and assessing the effect on compound binding and fibril dissolution. This would provide mechanistic support for the role of the binding pocket in compound action against fibril formation.

We appreciate the reviewer's insightful suggestion. We agree that site-directed mutagenesis of residues within the cavity, such as glycine or isoleucine, could provide mechanistic insight into the functional role of the binding pocket. However, introducing mutations at these positions may perturb the overall fibril architecture, complicating the interpretation of binding-specific effects. As our study focused on structural and biophysical characterization of anle138b binding, mutational validation was beyond its current scope. Nonetheless, we consider this a promising direction for future work to further elucidate the contribution of individual residues to compound binding and fibril modulation.

(7) The conclusion that Anle138b competes with ThT for binding may be premature. ThT fluorescence can be influenced by multiple factors, including fibril structure and accessibility. To confirm direct competition, the authors could fix the compound concentration and titrate increasing amounts of ThT to observe potential displacement effects. Such data would help validate this claim.

We appreciate the reviewer's valuable comments. In response, we have revised the relevant section to present a more cautious interpretation of the observed ThT fluorescence reduction. As the primary focus of this study was to characterize the interaction between anle138b and A β ₄₀ fibrils, the ThT signal decrease was treated

as a secondary observation. Similar reductions have been reported in previous studies involving small molecules and amyloid fibrils (as cited in the manuscript) and are not unique to our system.

We acknowledge that reduced ThT fluorescence may reflect changes in fibril structure or dye accessibility following compound binding, rather than direct competition. The reviewer's suggestion to titrate ThT under fixed compound concentration is a sound experimental approach for clarifying this mechanism and represents an important direction for future studies.

The manuscript has been revised to reflect the reviewer's comments as follows (135-139, highlighted in blue):

Lines 134-140: The observed reduction in ThT fluorescence intensity in the post-treatment sample (the fibril treated with anle138b after their formation) is likely attributable to competitive binding between anle138b and ThT at shared or nearby fibril binding sites. Thus, the fluorescence intensity cannot be used as a measure of fibril quantity, and complementary readouts suggest that the fibril quantity has not decreased significantly (Fig. 1b, c, Supplementary Fig. 3b). This interpretation is consistent with previously reported interactions between small molecules and amyloid fibrils^{21,22}.

(8) In the ITC experiments, the compound is solubilized in DLPG. It is unclear, however, whether the A β fibrils used in these experiments (post-treatment) were also prepared in DLPG or in a different lipid or buffer system. Since pre- and post-treatment experiments involve different lipids (DMPG vs. DLPG), this could affect the comparability of binding data and should be clarified.

We thank the reviewer for this important comment. As addressed in our response to Reviewer comments 2, the pre-formed fibrils used in both the ITC experiments and the post-treatment condition were generated in the presence of DMPG liposomes, under the same conditions as the control fibrils. This clarification is also reflected in the revised manuscript (*lines 125–126, highlighted in blue*).

We note that this point overlaps with Reviewer #2's earlier comments.

DLPG vesicles were used solely as delivery vehicles for anle138b in the post-treatment and ITC experiments and were not involved in fibril formation. Importantly, no detectable interaction between DLPG and the fibrils was observed by ITC, ssNMR, or cryo-EM. These combined results confirm that the binding data most likely reflect direct interactions between anle138b and DMPG-based fibrils, without confounding effects from lipid composition.

(9) The authors apply a flat-bottomed positional restraint in their MD simulations. More explanation is needed regarding the purpose of this restraint and whether it might limit

the natural flexibility of the compound-fibril interaction. Could this influence the binding mode or energetics observed?

Since Reviewer 2 also commented on this, we added further explanation to the MD simulation methods section, with the relevant details highlighted in blue.

When the molecule is within the flat bottom of the positional potential, the potential imposes no force on the molecule. The protein atoms of all residues in or around the central cavity are also not restrained. We therefore expect no influence on the compound-fibril interaction. Consequently, if we restrict our consideration to the cavity binding modes, then there should be no influence on the observed binding modes. Similar arguments apply for energetics. Of course, we cannot compare the results to surface binding, nor do we propose to compute binding affinities.

In more detail, the flat-bottomed restraint keeps the ligand within a cylinder above the central cavity of the L1 between the two protofilaments of the L1 A β ₄₀ fibril, thereby preventing it from diffusing away, and increasing the probability that the molecule encounters the cavity opening. This is crucial for observing the (otherwise unsteered or unbiased) probing and binding of the ligand to and into the central cavity within reasonable simulation times. The cylinder is chosen large enough to allow the molecule to rotate and orient freely above the central cavity (radius cylinder ~ longest extension of the molecule, see Supplementary Figure 24). As mentioned, the flat-bottomed positional restraint does influence the observed binding mode insofar as it restricts the conformational space available to the ligand to a cylindrical volume. If other, perhaps more stable or transient, binding modes exist outside the restricted volume, they are not observed with this set of simulations alone. However, these simulations were necessary to complement the wider range of alternative anle138b poses observed in the simulations with multiple ligands without restraints, since the limited sampling without restraints is insufficient to investigate the slower on-rates associated with cavity binding. The goal of the simulations of truncated fibril models with flat-bottomed positional restraints is primarily to show the *principal/general* ability of the ligand to bind into the central cavity and to record the interatomic protein-ligand distances while doing so. See also reply to comments (4) of **Reviewer #2 on p. 8-10**.

Reviewer #4 (Remarks to the Author):

REVIEWER COMMENTS

Reviewer #2 (Remarks to the Author):

I appreciate the authors' responsiveness and clarity. However, their revisions leave a couple of issues that I believe still need to be addressed:

1. I was confused by the rationale that that DLPG was selected to reduce charge-based nonspecific interactions, given that DLPG and DMPG have the same head group and charge state. The subsequent parallel rationale based on membrane phase and fluidity, while speculative, is clearer and more plausible.

Thank you for your comment.

We agree that the explanation stating DLPG was selected due to differences in charge state may be confusing. This sentence has been removed from the manuscript. Instead, we now explain the use of DLPG based on its lower melting temperature ($T_m = -3\text{ °C}$), which maintains high membrane fluidity at 25 °C and facilitates the dispersion of hydrophobic compounds such as anle138b.

2. The initial version of the manuscript clearly explained that the DLPG is likely to be in a micelle-like state and is referred to as 'vesicles' only for convenience. This is an essential disclosure that has been lost in the revision, and I believe it needs to be added back to the manuscript.

In response to your comment, we have reinserted the explanation regarding the physical state of DLPG in the Methods section. DLPG is likely to adopt a micelle-like structure under the experimental conditions used, and the term 'vesicle' is used merely as a convenient descriptor to simplify the description of the system.

Both points you raised have been addressed in the Methods section, highlighted in purple for clarity.

For post-treatment fibril, DLPG lipid films containing anle138b were hydrated and sonicated for 15 minutes in 10 mM sodium phosphate buffer (pH 6.5), yielding lipid-based particles <10 nm in diameter to solubilize the hydrophobic compound. Although the exact structural identity of DLPG particles remains unclear, with their small size (<10 nm) and physicochemical properties suggesting they are more likely micelles, non-bilayer structures, or ultrasmall unilamellar vesicles⁸¹, we refer to them as vesicles throughout this work for clarity. For post-treatment conditions, DLPG vesicles were used to deliver anle138b to pre-formed L1 A β ₄₀ fibrils. DLPG was selected over DMPG due to its lower melting temperature ($T_m = -3$ °C), which ensures high membrane fluidity at 25 °C and facilitates dispersion of hydrophobic molecules such as anle138b. To minimize co-sedimentation artifacts during centrifugation, small unilamellar DLPG vesicles were used. Control ITC experiments confirmed that DLPG vesicles alone exhibited no measurable binding enthalpy with L1 A β ₄₀ fibrils, supporting their role as a passive solubilizing vehicle for anle138b.